# Corruption Robust Active Learning

**Yifang Chen, Simon S. Du, Kevin Jamieson**
Paul G. Allen School of Computer Science & Engineering
University of Washington, Seattle,WA
{yifangc, ssdu, jamieson }@cs.washington.edu

## Abstract

We conduct theoretical studies on streaming-based active learning for binary classification under unknown adversarial label corruptions. In this setting, every time before the learner observes a sample, the adversary decides whether to corrupt the label or not. First, we show that, in a benign corruption setting (which includes the misspecification setting as a special case), with a slight enlargement on the hypothesis elimination threshold, the classical RobustCAL framework can (surprisingly) achieve nearly the same label complexity guarantee as in the non-corrupted setting. However, this algorithm can fail in the general corruption setting. To resolve this drawback, we propose a new algorithm which is provably correct without any assumptions on the presence of corruptions. Furthermore, this algorithm enjoys the minimax label complexity in the non-corrupted setting (which is achieved by RobustCAL) and only requires $\tilde{\mathcal{O}}(C_{\text{total}})$ additional labels in the corrupted setting to achieve $\mathcal{O}(\varepsilon + \frac{C_{\text{total}}}{n})$, where $\varepsilon$ is the target accuracy, $C_{\text{total}}$ is the total number of corruptions and $n$ is the total number of unlabeled samples.

## 1 Introduction

An active learning algorithm for binary classification aims to obtain the best hypothesis (classifier) from some given hypothesis set while requesting as few labels as possible. Under some favorable conditions, active learning algorithms can require exponentially fewer labels then passive, random sampling [Hanneke, 2014]. Active learning is ideally suited for applications where large datasets are required for accurate inference, but the cost of paying human annotators to label a dataset is prohibitively large [Joshi et al., 2009, Yang et al., 2015, Beluch et al., 2018]. A bit more formally, for an example space $\mathcal{X}$ (such as a set of images) and label space $\{0, 1\}$ (like whether the image contains a human-made object or not), let $\mathcal{H}$ be a hypothesis class such that for each $h \in \mathcal{H}$, we have $h : \mathcal{X} \to \{0, 1\}$. After a certain number of labels are requested, the learner will output a target hypothesis $h_{\text{out}} \in \mathcal{H}$. In this paper, we consider the streaming setting where at each time $t$ nature reveals $x_t \sim \mathcal{D}_X$ and an active learning algorithm must make the real-time decision on whether to request the corresponding label $y_t$ or not. Such a streaming setting of active learning is frequently encountered in online environments such as learning a spam filter or fraud detection (i.e., mark as spam/fraudulent and do not request the label, or send to inbox/expert to obtain a label).

This paper is interested in a setting when the requested label $y_t$ is potentially *corrupted* by an adversary. That is, when requesting the label for some example $x_t \in \mathcal{X}$, if uncorrupted the learner will receive a label drawn according to the "true" conditional label distribution, but if corrupted, the learner will receive a label drawn from an *arbitrary* distribution decided by an adversary. This setting is challenging because the learner has no a priori knowledge of when or how many corruptions will occur. And if the learner is collecting data adaptively, he may easily be misled into becoming confident in an incorrect belief, collect data based on that belief, and never recover to output an accurate classifier even if the adversary eventually stops serving corrupted labels later. This greatly contrasts with the passive setting (when all labels are observed) where as long as the number of

35th Conference on Neural Information Processing Systems (NeurIPS 2021).

corrupts grows sub-linearly over time, the effect of the corruptions will fade and the empirical risk minimizer will converge to an accurate classifier with respect to the uncorrupted labels.

The source of corruptions can come from automatic labeling, non-expert labeling, and, mostly severely, adaptive data poisoning adversaries. Particularly, with the rise of crowdsourcing, it is increasingly feasible for such malicious labelers to enter the system (Miller et al. [2014], Awasthi et al. [2014]). There have been many prior works that consider robust *offline* training using corrupted labels (i.e., the passive setting) [Hendrycks et al., 2018, Yu et al., 2019]. Correspondingly, related corruption settings have also been considered in online learning [Gupta et al., 2019, Zimmert and Seldin, 2019, Wei et al., 2020] and reinforcement learning [Lykouris et al., 2020, Chen et al., 2021a, Wei et al., 2021]. However, there is a striking lack of such literature in the active learning area. Existing disagreement-based active learning algorithms nearly achieve the minimax label complexity for a given target accuracy when labels are trusted [Hanneke, 2014], but they fail to deal with the case where the labels are potentially corrupted.

**Our contributions:**    In this paper, we study active learning in the agnostic, streaming setting where an unknown number of labels are potentially corrupted by an adversary. We begin with the performance of existing baseline algorithms.

- Firstly, we analyze the performance of empirical risk minimization (ERM) for the passive setting where all labels are observed, which will output an $\left(\varepsilon + \frac{R^* C_{\text{total}}}{n}\right)$-optimal hypothesis as long as $n \gtrsim \frac{1}{\epsilon} + \frac{R^*}{\varepsilon^2}$, where $R^*$ is the risk of best hypothesis. This result serves as a benchmark for the following active learning results (Section 3).

If we assume that the disagreement coefficient, a quantity that characterizes the sample complexity of active learning algorithms [Hanneke, 2014], is some constant, then we obtain the following results for active learning.

- Secondly, we analyze the performance of a standard active learning algorithm called Robust-CAL [Balcan et al., 2009, Dasgupta et al., 2007, Hanneke, 2014] under a benign assumption on the corruptions (misspecification model is a special case under that assumption).We show that, by slightly enlarging the hypothesis elimintion threshold, this algorithm can achieve almost the same label complexity as in the non-corrupted setting. That is, the algorithm will output an $\mathcal{O}(\varepsilon + \frac{R^* C_{\text{total}}}{n})$-optimal hypothesis as long as $n \gtrsim \frac{R^*}{\varepsilon^2} + \frac{1}{\varepsilon}$ with at most $\widetilde{\mathcal{O}}\left(R^* n + \log(n)\right)$ number of labels (Section 4).

- Finally and most importantly, in the general corruption case without any assumptions on how corruptions allocated, we propose a new algorithm that matches RobustCAL in the non-corrupted case and only requires $\widetilde{\mathcal{O}}(C_{\text{total}})$ additional labels in the corrupted setting. That is, the algorithm will output an $\left(\varepsilon + \frac{C_{\text{total}}}{n}\right)$-optimal hypothesis as long as $n \gtrsim \frac{1}{\varepsilon^2}$ with at most $\widetilde{\mathcal{O}}\left((R^*)^2 n + \log(n) + C_{\text{total}}\right)$ number of labels. Besides, this algorithm also enjoys an improved bound under a benign assumption on the corruptions. That is, the algorithm will output an $\left(\varepsilon + \frac{R^* C_{\text{total}}}{n}\right)$-optimal hypothesis with at most $\widetilde{\mathcal{O}}\left((R^*)^2 n + \log(n) + R^* C_{\text{total}}\right)$ number of labels (Section 5)

Note that $C_{\text{total}}$ can be regarded as a fixed budget or can be increasing with incoming samples. In the latter case, $C_{\text{total}}$ in the second and third result can be different since they may require different $n$. Detailed comparison between these two results will be discussed in corresponding sections.

**Related work:**    For nearly as long as researchers have studied how well classifiers generalize beyond their performance on a finite labelled dataset, they have also been trying to understand how to minimize the potentially expensive labeling burden. Consequently, the field of active learning that aims to learn a classifier using as few annotated labels as possible by selecting examples sequentially is also somewhat mature [Settles, 2011, Hanneke, 2014]. Here, we focus on just the agnostic, streaming setting where there is no relationship assumed a priori between the hypothesis class and the example-label pairs provided by Nature. More than a decade ago, a landmark algorithm we call RobustCAL was developed for the agnostic, streaming setting and analyzed by a number of authors that obtains nearly minimax performance [Balcan et al., 2009, Dasgupta et al., 2007, Hanneke, 2014].

The performance of RobustCAL is characterized by a quantity known as the disagreement coefficient that can be large, but in many favorable situations can be bounded by a constant $\theta^*$ which we assume is the case here. In particular, for any $\epsilon > 0$, once Nature has offered RobustCAL $n$ unlabelled samples, RobustCAL promises to return a classifier with error at most $\sqrt{R^* \log(|\mathcal{H}|)/n} + \log(|\mathcal{H}|)/n$ and requests just $nR^* + \theta^*(\sqrt{nR^* \log(|\mathcal{H}|)} + \log(|\mathcal{H}|))$ labels with high probability. Said another way, RobustCAL returns an $\epsilon$-good classifier after requesting just $\theta^*((R^*)^2/\epsilon^2 + \log(1/\epsilon))$ labels. If $\theta$ is treated as an absolute constant, this label complexity is minimax opitmal [Hanneke, 2014]. While there exist algorithms with other favorable properties and superior performance under special distributional assumptions (c.f., Zhang and Chaudhuri [2014], Koltchinskii [2010], Balcan et al. [2007], Balcan and Long [2013], Huang et al. [2015]), we use RobustCAL as our benchmark in the uncorrupted setting. We note that since RobustCAL is computationally inefficient for many classifier classes of interest, a number of works have addressed the issue at the cost of a higher label sample complexity [Beygelzimer et al., 2009, 2010, Hsu, 2010, Krishnamurthy et al., 2017] or higher unlabeled sample complexity [Huang et al., 2015]. Our own work, like RobustCAL, is also not computationally efficient but could benefit from the ideas in these works as well.

To the best of our knowledge, there are few works that address the non-IID active learning setting, such as the corrupted setting of this paper. Nevertheless, Miller et al. [2014] describes the need for robust active learning algorithms and the many potential attack models. While some applied works have proposed heuristics for active learning algorithms that are robust to an adversary [Deng et al., 2018, Pi et al., 2016], we are not aware of any that are provably robust in the sense defined in this paper. Active learning in crowd-sourcing settings where labels are provided by a pool of a varying quality of annotators, some active learning algorithms have attempted to avoid and down-weight poorly performing annotators, but these models are more stochastic than adversarial [Khetan and Oh, 2016]. The problem of selective sampling or online domain adaptation studies the setting where $P(Y_t = 1 | X_t = x)$ remains fixed, but $P(X_t = x)$ drifts and the active learner aims to compete with the best online predictor that observes all labels [Yang, 2011, Dekel et al., 2012, Hanneke and Yang, 2021, Chen et al., 2021b]. Another relevant line of work considers the case where the distribution of the examples drifts over time (i.e., $P(X_t = x)$) [Rai et al., 2010] or the label proportions have changed (i.e, $P(Y_t = 1)$) [Zhao et al., 2021], but the learner is aware of the time when the change has occurred and needs to adapt. These setting are incomparable to our own.

Despite the limited literature in active learning, there have been many existing corruption-related works in the related problem areas of multi-arm bandits (MAB), linear bandits and episodic reinforcement learning. To be specific, for MAB, Gupta et al. [2019] achieves $\tilde{\mathcal{O}}(\sum_{a \neq a^*} \frac{1}{\Delta_a} + KC)$ by adopting a sampling strategy based on the estimated gap instead of eliminating arms permanently. Our proposed algorithm is inspired by this "soft elimination" technique and requests labels based on the estimated gap of each hypothesis. Later, Zimmert and Seldin [2019] achieves a near-optimal result $\tilde{\mathcal{O}}\left(\sum_{a \neq a^*} \frac{1}{\Delta_a} + \sqrt{\sum_{a \neq a^*} \frac{C}{\Delta_a}}\right)$ in MAB by using Follow-the-Regularized Leader (FTRL) with Tsallis Entropy. How to apply the FTRL technique in active learning, however, remains an open problem. Besides MAB, Lee et al. [2021] achieves $\widetilde{\mathcal{O}}\left(\text{GapComplexity} + C\right)$ in stochastic linear bandits. We note that the linear bandits papers of Lee et al. [2021] and Camilleri et al. [2021] both leverage the Catoni estimator that we have repurposed for robust gap estimation in our algorithm. Finally, in the episodic reinforcement learning, Lykouris et al. [2020] achieves $\widetilde{\mathcal{O}}\left(C \cdot \text{GapComplexity} + C^2\right)$ in non-tabular RL and Chen et al. [2021a] achieves $\widetilde{\mathcal{O}}\left(\text{PolicyGapComplexity} + C^2\right)$ in tabular RL. Very recently, Wei et al. [2021] obtains an $\text{GapComplexity} + C$ bounds for linear MDP.

## 2   Preliminaries

**General protocol:**   A hypothesis class $\mathcal{H}$ is given to the learner such that for each $h \in \mathcal{H}$ we have $h : \mathcal{X} \to \{0, 1\}$. Before the start of the game, Nature will draw $n$ unlabeled samples in total. At each time $t \in \{1, \ldots, n\}$, nature draws $(x_t, y_t) \in \mathcal{X} \times \{0, 1\}$ independently from a joint distribution $\mathcal{D}_t$, the learner observes just $x_t$ and chooses whether to request $y_t$ or not. Note that in this paper, we assume $\mathcal{X}$ is countable, but it can be directly extended to uncountable case. Next, We denote the expected risk of a classifier $h \in \mathcal{H}$ under any distribution $\mathcal{D}$ as $R_{\mathcal{D}}(h) = \mathbb{E}_{x, y \sim \mathcal{D}} (\mathbf{1}\{h(x) \neq y\})$, the marginalized distribution of $x$ as $\nu$ and probability of $y = 1$ given $x$ and $\mathcal{D}$ as $\eta^x$. Finally we define $\rho_{\mathcal{D}}(h, h') = \mathbb{E}_{x \sim \nu} \mathbf{1}\{h(x) \neq h'(x)\}$.

**Uncorrupted model:** In the traditional uncorrupted setting, there exists a fixed underlying distribution $\mathcal{D}_*$ where each $(x_t, y_t)$ is drawn from this i.i.d distribution. Correspondingly, we define the marginalized distribution of $x$ as $\nu_*$ and probability of $y = 1$ given $x$ and $\mathcal{D}_*$ as $\eta_*^x$.

**Oblivious and non-oblivious adversary model:** In the corrupted setting, the label at time $t$ is corrupted if $(x_t, y_t)$ is drawn from some corrupted distribution $\mathcal{D}_t$ that differs from the base $\mathcal{D}_*$. At the start of the game, an oblivious adversary will choose a sequence of functions $\eta_t^x : \mathcal{X} \rightarrow [0, 1]$ for all $t \in \{1, \ldots, n\}$. The corruption level at time $t$ is measured as

$$c_t = \max_{x \in \mathcal{X}} |\eta_*^x - \eta_t^x|,$$

and the amount of corruptions during any time interval $\mathcal{I}$ as $C_{\mathcal{I}} = \sum_{t \in \mathcal{I}} c_t$. Correspondingly, we define $C_{\text{total}} = C_{[0,n]}$. Then, Nature draws $x_t \sim \nu_*$ for each $t \in \{1, \ldots, n\}$ so that each $x_t$ is independent of whether $y_t$ was potentially corrupted or not.

One notable case case of the oblivious model is the $\gamma$-misspecification model. In the binary classification setting, it is equivalent to

$$\eta_t^x = (1 - \gamma)\eta_*^x + \gamma \tilde{\eta}_t^x, \forall x, t.$$

where $\tilde{\eta}_t^x$ can be any arbitrary probability. Such label contamination model can be regarded a special case of corruption where for each $t$,

$$c_t = \max_x |\eta_t^x - \eta_*^x| = \gamma \max_x |\eta_*^x - \tilde{\eta}^x|$$

Moreover, our main algorithm actually works for the non-oblivious adversary. In this more challenging case, each time $t$, the adversary adaptively decides $\eta_t^x$ before seeing actual $x_t$, based on all the previous history.

**Other notations:** For convenience, we denote $R_{\mathcal{D}_t}(h)$ as $R_t(h)$, $R_{\mathcal{D}_*}(h)$ as $R_*(h)$, $\rho_{\mathcal{D}_t}(h, h') = \rho_t(h, h')$ and $\rho_{\mathcal{D}_*}(h, h') = \rho_*(h, h')$. We also define an average expected risk that will be used a lot in our analysis, $\bar{R}_{\mathcal{I}}(h) = \frac{1}{|\mathcal{I}|} \sum_{t \in \mathcal{I}} R_t(h)$. In addition, we define $h^* = \arg\min R_*(h)$, $R^* = R_*(h^*)$ and the gap of the suboptimal classifier $h$ as $\Delta_h = R_*(h) - R^*$.

**Disagreement coefficient:** For some hypothesis class $\mathcal{H}$ and subset $V \subset \mathcal{H}$, the region of disagreement is defined as $\text{Dis}(V) = \{x \in \mathcal{X} : \exists h, h' \in V \text{ s.t. } h(x) \neq h'(x)\}$, which is the set of unlabeled examples $x$ for which there are hypotheses in $V$ that disagree on how to label $x$. Correspondingly, the disagreement coefficient of $h^* \in \mathcal{H}$ with respect to a hypothesis class $\mathcal{H}$ and. distribution $\nu_*$ is defined as

$$\theta^*(r_0) = \sup_{r \geq r_0} \frac{\mathbb{P}_{x \sim \nu_*}(X \in \text{Dis}(B(h^*, r)))}{r}.$$

## 3 Passive Learning in the Corrupted Setting

We first analyze the performance of empirical risk minimization (ERM) for passive learning in the corrupted setting as a benchmark.

**Theorem 3.1** (Passive Learning). *After $n$ labeled samples, if $h_{out} = \arg\min_h \sum_{t=1}^n \mathbf{1}\{h(x_t) \neq y_t\}$ is the empirical risk minimizer, then with probability at least $1 - \delta$, we have*

$$R_*(h_{out}) - R^* \leq \frac{\log(|\mathcal{H}|/\delta)}{n} + \sqrt{\frac{8R^* \log(|\mathcal{H}|/\delta)}{n}} + \frac{8C_{total}}{n}R^* + \frac{5\log(|\mathcal{H}|/\delta)}{n}\frac{1}{(1 - \frac{4C_{total}}{n})^2},$$

*This implies that, as long as $C_{total}$ is small than some fraction of $n$, e.g., $C_{total} \leq \frac{n}{8}$, we can obtain $R_*(h_{out}) - R^* \leq \varepsilon + \frac{C_{total}}{n}R^*$ whenever*

$$n \geq \frac{2\log(|\mathcal{H}|/\delta)}{\varepsilon} + \frac{8R^* \log(|\mathcal{H}|/\delta)}{\varepsilon^2}$$

---

**Algorithm 1** RobustCAL (modified the elimination condition)

1: **Input:** confidence parameter $\delta$
2: **for** $t = 1, 2, \ldots, n$ **do**
3:     Nature reveals unlabeled data point $x_t$
4:     **if** $x_t \in \mathrm{Dis}(V_t)$ **then**
5:         Query $y_t$ and set $\hat{l}_t(h) = \mathbf{1}\{h(x_t) \neq y_t\}$ for all $h \in \mathcal{H}$
6:     **end if**
7:     **if** $\log(t) \in \mathbb{N}$ **then**
8:         Set $\hat{L}_t(h) = \frac{1}{t}\sum_{s \in t}\hat{l}_s(h)$ and $\hat{h}_t = \arg\min_{h \in V_t} \hat{L}_t(h)$
9:         Set $\hat{\rho}_t(h, h') = \frac{1}{t}\sum_t \mathbf{1}\{h(x_t) \neq h'(x_t)\}$ and $\beta_t = \log(3\log(t)|\mathcal{H}|^2/\delta)$
10:       Set $V_{t+1} = \left\{ h \in V_{\log(t)} : \hat{L}_t(h) - \hat{L}_t(\hat{h}_t) \leq \sqrt{\frac{2\beta_t\hat{\rho}_t(h,\hat{h}_t)}{t}} + \frac{3\beta_t}{2t} + \frac{1}{2}\hat{\rho}_t(h, \hat{h}_t) \right\}$
11:     **else**
12:         $V_{t+1} = V_t, \beta_{t+1} = \beta_t$
13:     **end if**
14: **end for**
15: **Output:** $\arg\min_{h \in V_t} \hat{L}_t(h)$

---

**Proof Sketch**     By using Bernstein inequality and the definition of corruptions, we can get

$$R_*(h_{\mathrm{out}}) - R^*$$

$$\leq \frac{4C_{\mathrm{total}}}{n}\max\{R_*(h_{\mathrm{out}}) - R^*, 2R^*\} + \sqrt{\frac{4\log(|\mathcal{H}|/\delta)\max\{R_*(h_{\mathrm{out}}) - R^*, 2R^*\}}{n}} + \frac{\log(|\mathcal{H}|/\delta)}{n}$$

Then we can directly get the result by solving this inequality. We postpone the details into Appendix B.

In addition to this result providing a benchmark, this passive learning result also inspires our analysis of RobustCAL in the corrupted setting as we will show in the next section.

## 4   Robust CAL in the Corrupted Setting

We restate the classical RobustCAL [Balcan et al., 2009, Dasgupta et al., 2007, Hanneke, 2014] in Algorithm 1 with slightly enlargement on the confidence threshold used in the elimination condition (Line 10). The additional term $\frac{1}{2}\hat{\rho}_t(h, \hat{h}_t)$ ensures robustness because each $(R_t(h) - R_t(h'))$ will be corrupted at most $2\rho_*(h, h')c_t$. In the theorem below we show that, it can achieve the similar label complexity result as in the non-corrupted setting as long as the growth rate of corruptions is at most in a certain fraction of number of unlabeled samples.

**Theorem 4.1.** *Suppose the* $C_{[0,t]} \leq \frac{t}{8}$ *for all* $t \in \{\log(t) = \mathbb{N}\}$*, for example, the* $(1/8)$-*misspecification model. Then with high probability as least* $1 - \delta$*, for any* $n \geq (\frac{8R^*}{\varepsilon^2} + \frac{22}{\varepsilon})\log(\log(n)|\mathcal{H}|^2/\delta)$*, we have* $R_{h_{out}} - R^* \leq \varepsilon + \mathcal{O}(\frac{R^*C_{total}}{n})$ *with label complexity at most*

$$\mathcal{O}\left(\theta^*(14R^* + 120\frac{\log(\log(n)|\mathcal{H}|/\delta)}{n})\log(\log(n)|\mathcal{H}|^2/\delta)\left(R^*n + \log(n)\right)\right)$$

*Remark* 4.1. In Appendix C.2, we show the necessity of enlarging the threshold in line 10 from the original

$$V_{t+1} = \left\{ h \in V_{\log(t)} : \hat{L}_t(h) - \hat{L}_t(\hat{h}_t) \leq o\left(\sqrt{\frac{2\beta_t\hat{\rho}_t(h, \hat{h}_t)}{t}} + \frac{\beta_t}{t}\right) \right\}.$$

by giving an counter-example. The counter-example shows that, when $R^* \gg 0$, the best hypothesis will be eliminated under the original condition even the "$C_{[0,t]} \leq \frac{t}{8}$ for all $t \in \{\log(t) = \mathbb{N}\}$" assumption is satisfied.

**Proof Sketch**     For correctness, it is easy to show by Bernstein inequality. For the sample complexity, Theorem 3.1 implies that, for any interval $[0, t]$, as long as $C_{[0,t]} \leq \frac{t}{8}$, the learner can always identify

hypothesis which are $\mathcal{O}(R^* + \frac{1}{n})$-optimal. Therefore, we get the probability of query as

$$\mathbb{P}\left(x_{t+1} \in \text{Dis}(V_{t+1})\right) \leq \mathbb{P}\left(\exists h \in V_{t+1} : h(x_t) \neq h^*(x_t), \Delta_h \leq \mathcal{O}\left(R^* + \frac{\beta_t}{t}\right)\right)$$

Then by standard analysis we can connect this disagreement probability with the disagreement coefficient to get the final bound. One thing to note is that, at the first glance $\hat{\rho}_t(h, \hat{h}_t)$ might be much larger than the other two terms since it can goes to 1, which possibly renders a worse label complexity. Here we give an intuitive explanation on why this threshold is fine: If $\hat{\rho}_t(h, \hat{h}_t)$ is close to the $|R(h) - R(\hat{h}_t)|$, then we can achieve the inequality above by using some self-bounding techniques. If $\hat{\rho}_t(h, \hat{h}_t)$ is close to the $R^*$, then we can directly get some $R^*$-dependent term in the target bound. The full proof is deferred to Appendix C.1.

**Comparison between the modified RobustCal and passive learning:** Assume disagreement coefficient is a constant. In the non-corrupted case, the algorithm achieves the same performance guarantee as the vanilla Robust CAL. In the corrupted case, we still get the same accuracy as in Theorem 3.1 with at most $\widetilde{\mathcal{O}}(R^*n + \log(n))$ number of labels, which is the same as the non-corrupted case.

**Discussion on the "$C_{[0,t]} \leq \frac{t}{8}$ for all the $\{t \mid \log(t) \in \mathbb{N}\}$" condition:** This condition can be reduced to the $(1/8)$-misspecification model as defined in Section 2 since $C_{\mathcal{I}} \leq \frac{|\mathcal{I}|}{8}$ for any $\mathcal{I}$. But this condition does not contain the case where an adaptive poisoning adversary corrupts all the labels at the earlier stage and stop corrupting later, which still ensures the small total amount of corruptions, but will clearly mislead the algorithm to delete a true best hypothesis $h^*$. *In Section 5, we will show a more general result that applies to scenarios beyond $C_{[0,t]} \leq \frac{t}{8}$.*

## 5 Main algorithm - CALruption

### 5.1 Algorithm

In this section we describe our new algorithm, CALruption. The pseudo-code is listed in Algorithm 2. Our previous analysis showed that in the agnostic setting the classical RobustCAL may permanently eliminate the best hypothesis due to the presence of corruptions. To fix this problem, in our CALruption algorithm, the learner never makes a "hard" decision to eliminate any hypothesis. Instead, it assigns different query probability to each $x$ based on the estimated gap for each hypothesis as shown in line 4 and 5, which can be regarded as "soft elimination". With this step, the key question becomes how to connect the estimated gaps with the query probability $q_l^x$.

We adopt the idea from the BARBAR algorithm proposed by Gupta et al. [2019] which was originally designed for multi-armed bandits (MAB). Instead of permanently eliminating a hypothesis, the learner will continue pulling each arm with a certain probability defined by its estimated gap. However, the original BARBAR algorithm is mainly focused on estimating the reward of each individual arm. This aligns with its MAB feedback structure, where only the information of the pulled arm will be gained at each time. In the active learning setting, we instead focus on the *difference* of the risks of different hypotheses, because each time we request a label, values of all the hypotheses will be updated. Therefore, we implement a more complicated strategy to calculate the query probability at the end of each epoch $l$, as shown from line 7 to line 13.

In line 7, we estimate the disagreement probability for each hypothesis pair $(h, h')$ with an empirical quantity that upper bounds the expectation. In line 8, instead of estimating the value of each hypothesis, we estimate the gap between each hypothesis pair $(h, h')$, denoted as $W_l^{h,h'}$, by any $\delta$-robust estimator that satisfies eq. 1. One example of $\delta$-robust estimator is Catoni estimator [Lugosi and Mendelson, 2019]. Note that simple empirical estimator will lead to potentially rare but large variance, which has been discussed in Stochastic rounding section in Camilleri et al. [2021]. But what we truly care is the gap between any hypothesis $h$ and the best hypothesis $h^*$. Therefore, inspired by Camilleri et al. [2021], we construct such estimation by using $W_l^{h,h'}$ as shown in line 9 to 11. Finally, we divide the hypothesis set into several layers based on the estimated gap and set the query probability for each $x$ based on the hypothesis layers, as shown in line 12 and 13. For more detailed explanation on line 9-13, please refer to Appendix D.

**Algorithm 2** CALruption

1: **Initialize:** $\beta_3 = 2\log(\frac{3}{2}\lfloor\log(n)\rfloor|\mathcal{H}|^2/\delta), \beta_1 = 32*640\beta_3, \beta_2 = \frac{5}{32}, \epsilon_i = 2^{-i}, N_l = \beta_1\epsilon_l^{-2},$
$\hat{\Delta}_h^0 = 0, V_1^0 = \mathcal{Z}$ and $\tau_1 = 1, q_l^x = 1$ for all $x \in \mathcal{X}$
2: **for** $t = 1, 2, \ldots, n$ **do**
3:     Nature reveals unlabeled data point $x_t$
4:     Set $Q_t \sim \text{Ber}(q_l^x)$ and request $y_t$ if $Q_t = 1$.
5:     Set estimated loss for all $h \in \mathcal{H}$ as $\hat{\ell}_t(h) = \frac{\mathbf{1}\{h(x_t)\neq y_t\}}{q_l^x}Q_t$
6:     **if** $t = \tau_l + N_l - 1$ **then**
7:         Set $\hat{\rho}_l(h, h') = \frac{1}{N_l}\sum_{t\in\mathcal{I}_l}\mathbf{1}\{h(x_t) \neq h'(x_t)\}$ for all $h, h' \in \mathcal{H}$
8:         For each $(h, h')$, set $W_l^{h,h'} = \text{RobustEstimator}\left(\{\hat{\ell}_t(h) - \hat{\ell}_t(h')\}_{t\in\mathcal{I}_l}\right)$, which satisfies that, with probability at least $1 - \delta$,

$$|(\hat{R}_l(h) - \hat{R}_l(h')) - W_l^{h,h'}| \leq \sqrt{\frac{10\beta_3\hat{\rho}_l(h, h')}{N_l \min_{x\in\text{Dis}(h,h')} q_l^x}}, \tag{1}$$

        where $\hat{R}_l(h) = \frac{1}{|\mathcal{I}_l|}\sum_{t\in\mathcal{I}_l}\mathbb{E}_{y\sim\text{Ber}(\eta_t^{x_t})}[\mathbf{1}\{h(x_t) \neq y\}]$.
9:         Set $\hat{\mathcal{D}}_l = \arg\min_{\mathcal{D}}\max_{h,h'\in\mathcal{H}}(R_{\mathcal{D}}(h) - R_{\mathcal{D}}(h') - W_l^{h,h'})\sqrt{\frac{\min_{x\in\text{Dis}(h,h')} q_l^x}{\hat{\rho}_l(h,h')}}$
10:        Set $\hat{h}_*^l = \arg\min_{h\in\mathcal{H}}\left(R_{\hat{\mathcal{D}}_l}(h) + \beta_2\hat{\Delta}_h^{l-1}\right)$
11:        Set $\hat{\Delta}_h^l = \max\left\{\epsilon_l, R_{\hat{\mathcal{D}}_l}(h) - \left(R_{\hat{\mathcal{D}}_l}(\hat{h}_*^l) + \beta_2\hat{\Delta}_{\hat{h}_*^l}^{l-1}\right)\right\}$
12:        Construct $V_{l+1}^i$ for all $i = 0, 1, 2, \ldots, l$, such that,

$$\hat{\Delta}_h^l \leq \epsilon_i, \forall h \in V_{l+1}^i \quad\text{and}\quad \hat{\Delta}_h^l > \epsilon_i, \forall h \notin V_{l+1}^i$$

        Therefore, $V_{l+1}^l \subset V_{l+1}^{l-1} \subset \ldots \subset V_{l+1}^0$
13:        Calculate the query probability $q_l^x$ for each $x$ as follows

$$\mathcal{Z}(x) = \{(h, h') \in \mathcal{H} \mid x \in \text{Dis}(\{h, h'\})\}$$
$$k(h, h', l+1) = \max\{i \mid h, h' \in V_{l+1}^i\}$$
$$q_{l+1}^x = \max_{(h,h')\in\mathcal{Z}(x)}\frac{\beta_1\hat{\rho}_l(h, h')}{N_{l+1}}\epsilon_{k(h,h',l+1)}^{-2}$$

14:        Set $\tau_{l+1} = \tau_l + N_l$ and denote the epoch $l$ as $\mathcal{I} = [\tau_l, \tau_{l+1} - 1]$. Set $l \leftarrow l + 1$, go to the next epoch
15:     **end if**
16: **end for**
17: **Output:** $h \in V_l^{l-1}$

---

*Remark* 5.1. In Line 9, instead of estimating over all possible distribution $\mathcal{D}$, we actually just need to estimate $\eta_*^x$ for all $x \in \{x_t\}_{t\in\mathcal{I}_l}$ and set the corresponding $x$ distribution of $\mathcal{D}$ as the empirical distribution of $x$ inside $\mathcal{I}_l$.

**Theorem 5.1** (CALruption). *With $n \geq 72\varepsilon^{-2}\beta_1$ number of unlabeled samples, with probability at least $1 - \delta$ we can get an $h_{out}$ satisfying*

$$R_*(h_{out}) - R^* \leq \varepsilon + 24\frac{\overline{C}_{total}}{n},$$

*with label complexity as most*

$$\mathcal{O}\left(\theta^*(R^* + 3\sqrt{\frac{\beta_1}{n}} + \frac{64\overline{C}_{total}}{n})\log(\log(n)|\mathcal{H}|^2/\delta)\left((R^*)^2n + \log(n)(1 + \overline{C}_{total})\right)\right)$$

*where $\overline{C}_{total} = \sum_{l=1}^{\lfloor\log_4(n/\beta_1)\rfloor} C_{epoch\,l}\left(R^*\mathbf{1}\{\frac{C_{epoch\,l}}{N_l} \leq \frac{1}{32}\} + \mathbf{1}\{\frac{C_{epoch\,l}}{N_l} > \frac{1}{32}\}\right)$ and $\beta_1 = 16 * 640\log(\frac{3}{2}\lfloor\log(n)\rfloor|\mathcal{H}|^2/\delta)$. Note that epoch $l$ is prescheduled and not algorithm-dependent.*

**Corollary 5.1.** *Suppose the corruptions satisfy $\frac{C_{epoch\,l}}{N_l} \leq \frac{1}{32}$ for all epochs, for example, the $(1/32)$-misspecification case, then for any $n \geq 72\varepsilon^{-2}\beta_1$ number of unlabeled samples, with probability at least $1 - \delta$ we can get a $h_{out}$ satisfying*

$$R_*(h_{out}) - R^* \leq \varepsilon + 24R^* \frac{C_{total}}{n},$$

*with label complexity as most*

$$\mathcal{O}\left(\theta^*(R^* + 3\sqrt{\frac{R^*\beta_1}{n}} + \frac{64R^*C_{total}}{n})\log(\log(n)|\mathcal{H}|/\delta)\left((R^*)^2 n + (R^*C_{total} + 1)\log(n)\right)\right)$$

**Comparison with passive learning and the Calruption:** Consider the case where $\theta^*(\cdot)$ is of lower order like a constant. The Corollary 5.1 shows that, when $\frac{C_{epoch\,l}}{N_l} \leq \frac{1}{32}$ for all epochs, our algorithm achieves a similar accuracy $\mathcal{O}\left(\varepsilon + \frac{R^*C_{total}}{n}\right)$ as in the passive learning case, while only requiring $\widetilde{\mathcal{O}}\left((R^*)^2 n + \log(n)(1 + R^*C_{total})\right)$ number of labels, for $n \gtrsim \frac{1}{\varepsilon^2}$. So if we set $n = \widetilde{\mathcal{O}}(\frac{1}{\varepsilon^2})$, then the label complexity becomes $\widetilde{\mathcal{O}}\left(\frac{(R^*)^2}{\varepsilon^2} + \log(1/\varepsilon)(1 + R^*C_{total})\right)$, which matches the minimax label complexity in the non-corrupted case.

Going beyond the $\frac{C_{epoch\,l}}{N_l} \leq \frac{1}{32}$ constraint, the general Theorem 5.1 shows that, for $n \gtrsim \frac{1}{\varepsilon^2}$, our algorithm achieves an accuracy $\mathcal{O}\left(\varepsilon + \frac{C_{total}}{n}\right)$ while only requiring $\widetilde{\mathcal{O}}((R^*)^2 n + \log(n) + C_{total})$ number of labels no matter how corruptions are allocated. When $R^*$ is some constant, this result becomes similar to the Corollary 5.1. Moreover, we will argue that upper bound $\overline{C}_{total}$ by $C_{total}$ is loose and in many case $\overline{C}_{total}$ will be close to $R^*C_{total}$ instead of $C_{total}$. We show one example in the paragraph below.

**When is Calruption better than modified Robust CAL?** Consider the case where the adversary fully corrupts some early epoch and then performs corruptions satisfying $\frac{C_{epoch\,l}}{N_l} \leq \frac{1}{32}$ for rest epochs. Then the modified Robust CAL will mistakenly eliminate $h^*$ so it can never achieve target result when $\varepsilon < \min_{h \in \mathcal{H}} \Delta_h$ while Calruption can surely output the correct hypothesis. Moreover, according to Theorem 5.1, since the total amount of early stage corruptions are small, so here $\overline{C}_{total}$ is close to $R^*C_{total}$, which implies a similar result as in Corrolary 5.1.

**When is Calruption worse then modified Robust CAL ?** Consider the case where the total amount of corruption is, instead of fixed, increasing with incoming unlabeled samples, for example, the misspecification case. Then $C_{total}$ in modified Robust CAL can be $\mathcal{O}(\frac{R^*}{\varepsilon^2} + \frac{1}{\varepsilon})$ while $C_{total}$ in CALruption can goes to $\mathcal{O}(\frac{1}{\varepsilon^2})$. Such gap comes from the extra unlabeled sample complexity, which we discuss in the paragraph below.

**Discussion on the extra unlabeled samples complexity:** We note that we require a larger number of unlabeled data than ERM in the passive learning setting. Here we explain the reason. Consider the version spaces $V_l^{l-1}$ for any fixed epoch $l$. In the non-corrupted setting, this version space serves the similar purpose as the active hypothesis set in Robust CAL. In Robust CAL, its elimination threshold is about $\widetilde{\mathcal{O}}\left(\sqrt{\frac{\rho_*(h,h')}{t}} + \frac{1}{t}\right)$ (or $\widetilde{\mathcal{O}}\left(\rho_*(h,h') + \frac{1}{t}\right)$ in our modified version) while in our CALruption, the threshold is about $\widetilde{\mathcal{O}}\left(\sqrt{\frac{1}{t}}\right)$, which is more conservative than the Robust CAL and leads to the extra unlabeled sample complexity. The reason about being conservative here is that we need more samples to weaken the effects of corruptions on our estimation. Whether such extra unlabeled samples complexity is unavoidable remains an open problem.

## 5.2 Proof sketch for Theorem 5.1

Here we provide main steps of the proof and postpone details in Appendix E.

First we show a key lemma which guarantees the closeness between $\hat{\Delta}_h^l$ and $\Delta_h$ for all $l$ and $h$.

**Lemma 5.1** (Upper bound and lower bound for all estimation). *With probability at least $1 - \delta$, for all epoch $l$ and all $h \in \mathcal{H}$,*

$$\hat{\Delta}_h^l \leq 2\left(\Delta_h + \epsilon_l + g_l\right), \qquad\qquad \Delta_h \leq \frac{3}{2}\hat{\Delta}_h^l + \frac{3}{2}\epsilon_l + 3g_l,$$

*where $g_l = \frac{2}{\beta_1}\epsilon_l^2 \sum_{s=1}^{l} C_s \left(2R^*\mathbf{1}\left\{\frac{2C_{\mathcal{I}_s}}{N_s} \leq \frac{1}{16}\right\} + \mathbf{1}\left\{\frac{2C_{\mathcal{I}_s}}{N_s} > \frac{1}{16}\right\}\right).$*

Here the $g_l$ term implies that, as long as the total corruption is sublinear in $n$, the misleading effects on the gap estimations will fade when the number of unlabeled samples increasing.

Based on this lemma, we can directly get another useful lemma as follows.

**Lemma 5.2.** *For all epoch $l$ and layer $j$, we have $\max_{h \in V_l^j} \rho_*(h, h^*) \leq 2R^* + 3\epsilon_j + 3g_{l-1}$*

In the following we first deal with the correctness then then sample complexity.

**Correctness.** By Lemma 5.1, we have

$$\Delta_{h_{out}} \leq \frac{3}{2}\hat{\Delta}_{h_{out}}^{L-1} + \frac{3}{2}\epsilon_{L-1} + 3g_{L-1} \leq 6\sqrt{\frac{2\beta_1}{n}} + 24\frac{\bar{C}_{total}}{n}.$$

**Sample complexity.** For any $t \in \mathcal{I}_l$, recall that $q_l^x = \max_{(h,h')\in\mathcal{Z}(x)} \frac{\beta_1\hat{\rho}_{l-1}(h,h')}{N_l}\epsilon_{k(h,h',l)}^{-2}$, the probability of $x_t$ being queried ($Q_t = 1$) is

$$\mathbb{E}[Q_t] \leq 10\frac{\beta_1}{N_l}\sum_{x\in\mathcal{X}} P(x_t = x)\max_{h\in V_l^{j_l^x}}\rho_*(h,h^*)\epsilon_{j_l^x}^{-2} + 8\frac{\beta_1}{N_l}$$

$$\leq 10\frac{\beta_1}{N_l}\sum_{x\in\mathcal{X}} P(x_t = x)\left(2R^*\epsilon_{j_l^x}^{-2} + 3\epsilon_{j_l^x}^{-1} + 3g_{l-1}\epsilon_{j_l^x}^{-2}\right) + 8\frac{\beta_1}{N_l}$$

$$\leq 10\frac{\beta_1}{N_l}\sum_{i=0}^{l-1}\left(2R^*\epsilon_i^{-2} + 3\epsilon_i^{-1} + 3g_{l-1}\epsilon_i^{-2}\right)\mathbb{P}(x\in\mathrm{Dis}(V_l^i)) + 8\frac{\beta_1}{N_l}$$

Here $j_l^x$ is some arbitrary mapping from $\mathcal{X}$ to $[l]$, which is formally defined in detailed version in Appendix E.6. The first inequality comes from the closeness of estimated $\hat{\rho}_l(h,h')$ and the true $\rho_*(h,h')$, as well as some careful relaxation. The second inequality comes from Lemma 5.2.

Now we can use the standard techniques to upper bound $\mathbb{P}(x\in\mathrm{Dis}(V_l^i))$ as follows,

$$\mathbb{P}\left(\exists h \in V_l^i : h(x) \neq h^*(x)\right) \leq \mathbb{P}\left(\exists h \in \mathcal{H} : h(x) \neq h^*(x), \rho_*(h,h^*) \leq 2R^* + 3\epsilon_i + 3g_{l-1}\right)$$

$$\leq \theta^*(2R^* + 3\epsilon_i + g_{l-1})\left(2R^* + 3\epsilon_i + 3g_{l-1}\right)$$

where again the first inequality comes from Lemma 5.2. Again we postpone the full version into Appendix E.6.

Combining the above results with the fact that $g_l = \frac{2}{\beta_1}\epsilon_l^2\bar{C}_{l-1}$ and $\bar{C}_{l-1} \leq \sum_{s=1}^{l-1} C_{\mathcal{I}_s} \leq 2\beta_1\epsilon_{l-1}^{-2}$, we get the expected number of queries inside a complete epoch $l$ as,

$$\sum_{t\in\mathcal{I}_l}\mathbb{E}[Q_t] \leq 20\beta_1\theta^*(2R^* + 3\epsilon_{l-1} + g_{l-1}) * \left(4(R^*)^2\epsilon_l^{-2} + 12R^*\epsilon_l^{-1} + \frac{132}{\beta_1}\bar{C}_{l-1} + 10\right)$$

Finally, summing over all $L = \lceil\frac{1}{2}\log(n/\beta_1)\rceil$ number of epochs, for any $n$, we can get the target lable complexity.

# 6 Conclusion and future works

In this work we analyzed an existing active learning algorithm in the corruption setting, showed when it fails, and designed a new algorithm that resolve the drawback. Relative to RobustCAL,

our algorithm requires a larger number of unlabeled data. One natural question is to design a corruption robust algorithm which requires the same number of unlabeled data as RobustCAL in the non-corrupted setting. Another potential question is that, the $\mathcal{O}\left(\varepsilon + \frac{\overline{C}_{\text{total}}}{n}\right)$ accuracy from Algo. 2 in the general corruption case is generally worse than $\mathcal{O}\left(\varepsilon + \frac{R^* C_{\text{total}}}{n}\right)$ accuracy of passive learning. Although we state that these two bounds are close in many cases, a question is if there exists an alternative algorithm or analysis that will result a more smooth final bound to interpolate between different corruptions cases.

Finally, we believe it is possible to replace the Catoni estimator with naive importance sampling estimator and even further simplify the layered active hypothesis sets construction step by adopting Beygelzimer et al. [2010]'s idea. We plan to work on this direction in the future.

## Acknowledgements

SSD gratefully acknowledges funding from NSF Award's IIS-2110170 and DMS-2134106

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
