# Contents

# A  Lemmas related to corruption effects

Here we states some basic lemmas that will be used all over the proofs.

**Lemma A.1** (Corruption effects 1). *For any interval $\mathcal{I}$ and hypothesis $h$, we have*

$$\frac{1}{|\mathcal{I}|} \sum_{t \in \mathcal{I}} (R_t(h) - R_*(h)) \le \frac{C_\mathcal{I}}{|\mathcal{I}|}$$

*Proof.*

$$\frac{1}{|\mathcal{I}|} \sum_{t \in \mathcal{I}} (R_t(h) - R_*(h))$$

$$= \mathbb{E}_{x \sim \nu_*} \frac{1}{|\mathcal{I}|} \sum_{t \in \mathcal{I}} \left( \mathbb{E}_{y \sim \eta_t^x} \left[ \mathbf{1}\{h(x) \neq y\} \right] - \mathbb{E}_{y \sim \eta_*^x} \left[ \mathbf{1}\{h(x) \neq y\} \right] \right)$$

$$\leq \frac{1}{|\mathcal{I}|} \sum_{t \in \mathcal{I}} \max_{x \in \mathcal{X}} \left( \mathbb{E}_{y \sim \eta_t^x} \left[ \mathbf{1}\{h(x) \neq y\} \right] - \mathbb{E}_{y \sim \eta_*^x} \left[ \mathbf{1}\{h(x) \neq y\} \right] \right)$$

$$\leq \frac{1}{|\mathcal{I}|} \sum_{t \in \mathcal{I}} \max_{x \in \mathcal{X}} |\eta_t^x - \eta_*^x| \leq \frac{C_{\mathcal{I}}}{|\mathcal{I}|}$$

$\square$

**Lemma A.2** (Corruption effects 2). *For any interval $\mathcal{I}$ and hypothesis pair $h, h'$, we have*

$$\frac{1}{|\mathcal{I}|} \sum_{t \in \mathcal{I}} (R_t(h) - R_t(h')) - (R_*(h) - R_*(h')) \leq 2\rho_*(h, h') \frac{C_{\mathcal{I}}}{|\mathcal{I}|}$$

*Proof.*

$$\frac{1}{|\mathcal{I}|} \sum_{t \in \mathcal{I}} (R_t(h) - R_t(h')) - (R_*(h) - R_*(h'))$$

$$= \mathbb{E}_x \left[ \frac{1}{|\mathcal{I}|} \sum_{t \in \mathcal{I}} \left( \mathbb{E}_{t \sim \eta_t^x} \left[ \mathbf{1}\{h(x) \neq y\} - \mathbf{1}\{h'(x) \neq y\} \right] - \mathbb{E}_{t \sim \eta_*^x} \left[ \mathbf{1}\{h(x) \neq y\} - \mathbf{1}\{h'(x) \neq y\} \right] \right) \right]$$

$$= \mathbb{E}_x \left[ \frac{\mathbf{1}\{h(x) \neq h'(x)\}}{|\mathcal{I}|} \sum_{t \in \mathcal{I}} \left( \mathbb{E}_{t \sim \eta_t^x} \left[ \mathbf{1}\{h(x) \neq y\} - \mathbf{1}\{h'(x) \neq y\} \right] - \mathbb{E}_{t \sim \eta_*^x} \left[ \mathbf{1}\{h(x) \neq y\} - \mathbf{1}\{h'(x) \neq y\} \right] \right) \right]$$

$$\leq \rho_*(h, h') \left( \frac{2}{|\mathcal{I}|} \sum_{t \in \mathcal{I}} \max_{h \in \mathcal{H}} (R_t(h) - R_*(h)) \right)$$

$$\leq 2\rho_*(h, h') \frac{C_{\mathcal{I}}}{|\mathcal{I}|}$$

$\square$

# B   Analysis for Passive Learning: Proof of Theorem 3.1

With probability at least $1 - \delta$, we have for any $n$ samples,

$R_*(h_{\text{out}}) - R^*$

$\leq \left( (R_*(h_{\text{out}}) - R^*) - (\bar{R}_{[1,n]}(h_{\text{out}}) - \bar{R}_{[1,n]}(h^*)) \right) + \left( \bar{R}_{[1,n]}(h_{\text{out}}) - \bar{R}_{[1,n]}(h^*) \right)$

$\leq 2\frac{C_{\text{total}}}{n} \rho_*(h_{\text{out}}, h^*) + \left( \bar{R}_{[1,n]}(h_{\text{out}}) - \bar{R}_{[1,n]}(h^*) \right)$

$\leq 2\frac{C_{\text{total}}}{n} \rho_*(h_{\text{out}}, h^*) + \left( \hat{R}_{[1,n]}(h_{\text{out}}) - \hat{R}_{[1,n]}(h^*) \right) + \sqrt{\rho_*(h_{\text{out}}, h^*)\frac{4\log(|\mathcal{H}|/\delta)}{n}} + \frac{\log(|\mathcal{H}|/\delta)}{n}$

$\leq 4\frac{C_{\text{total}}}{n} \max\{R_*(h_{\text{out}}) - R^*, 2R^*\} + \sqrt{\max\{R_*(h_{\text{out}}) - R^*, 2R^*\}\frac{4\log(|\mathcal{H}|/\delta)}{n}} + \frac{\log(|\mathcal{H}|/\delta)}{n}$

where the second step can from our definition of corruptions and fact that $\nu_*$ is not corrupted (see Lemma A.2 for details), third inequality comes from the Bernstein inequality and the last inequality comes from the definition of $h_{\text{out}}$ and the fact $\rho_*(h, h') \leq 2\max\{R_*(h) - R^*, 2R^*\}$. Now if $2R^* \geq R_*(h_{\text{out}}) - R^*$, then we directly get the target result. Otherwise, by solving the quadratic inequality, we have

$$R_*(h_{\text{out}}) - R^* \leq \frac{5\log(|\mathcal{H}|/\delta)}{n} \frac{1}{(1 - \frac{4C_{\text{total}}}{n})^2}$$

## C  Analysis for Robust CAL

### C.1  Proof of Theorem 4.1

For convenient, for all subscripts $[0, t]$, we simply write as subscript $t$.

We first state a key lemma that is directly inspired by Theorem 3.1.

**Lemma C.1.** *For any $t$ that $\log(t) = \mathbb{N}$, under the assumption of this theorem, as long as $h^* \in V_t$, we have*

$$
\begin{aligned}
R_*(\hat{h}_t) - R^* &\leq \frac{22 \log(|\mathcal{H}|/\delta)}{t} + 4\frac{C_t}{n}R^* + \sqrt{R^* \frac{8 \log(|\mathcal{H}|/\delta)}{t}} \\
&\leq \frac{22 \log(|\mathcal{H}|/\delta)}{t} + \frac{R^*}{2} + \sqrt{R^* \frac{8 \log(|\mathcal{H}|/\delta)}{t}} \quad \text{(By assumption on $C_t$)} \\
&\leq \frac{26 \log(|\mathcal{H}|/\delta)}{t} + R^* \quad \text{(By the fact $\sqrt{AB} \leq \frac{A+B}{2}$)}
\end{aligned}
$$

*Proof.* With probability at least $1 - \delta$, by combine the same proof steps as in Theorem 3.1 and the fact that $\hat{R}_{[1,t]}(\hat{h}_t) - \hat{R}_{[1,t]}(h^*) = \hat{L}_t(\hat{h}_t) - \hat{L}_t(h^*) \leq 0$, we can get the similar inequality as follows

$$
R_*(\hat{h}_t) \leq 4\frac{C_t}{n}\max\{R_*(\hat{h}_t) - R^*, 2R^*\} + \sqrt{\max\{R_*(\hat{h}_t) - R^*, 2R^*\}\frac{4\log(|\mathcal{H}|/\delta)}{t}} + \frac{\log(|\mathcal{H}|/\delta)}{t}
$$

Then again by quadratic inequality and the assumption that $\frac{C_t}{t} \leq \frac{1}{8}$, we have

$$
R_*(\hat{h}_t) \leq \frac{22 \log(|\mathcal{H}|/\delta)}{t} + 4\frac{C_t}{n}R^* + \sqrt{R^* \frac{8 \log(|\mathcal{H}|/\delta)}{t}}
$$

$\square$

This lemma suggests that, as long as the corruptions are not significantly large. For example, in this theorem, $C_t \leq \frac{1}{8}t$. Then the learner can still easily identify the $\widetilde{\mathcal{O}}(\frac{1}{t} + R^*)$-optimal hypothesis even in the presence of corruptions. Therefore, we can guarantee that the best hypothesis always stay in active set $V_t$ after elimination. We show the detailed as follows.

Define $\mathcal{E}_1, \mathcal{E}_2$ as

$$
\mathcal{E}_1 := \left\{ \forall t \text{ that } \log(t) = \mathbb{N}, (\overline{R}_t(h) - \overline{R}_t(h')) - (\hat{R}_t(h) - \hat{R}_t(h')) \leq \sqrt{\frac{2\beta_t \hat{\rho}_t(h, h')}{t}} + \frac{\beta_t}{t} \right\}
$$

$$
\mathcal{E}_2 := \left\{ \forall t \text{ that } \log(t) = \mathbb{N}, (\overline{R}_t(h) - \overline{R}_t(h')) - (\hat{R}_t(h) - \hat{R}_t(h')) \leq \sqrt{\frac{2\beta_t \rho_*(h, h')}{t}} + \frac{\beta_t}{t} \right\}
$$

$$
\mathcal{E}_3 := \left\{ \forall t \text{ that } \log(t) = \mathbb{N}, |\rho_*(h, h') - \hat{\rho}_t(h, h')| \leq \sqrt{\frac{2\beta_t \hat{\rho}_t(h, h')}{t}} + \frac{\beta_t}{t} \right\}
$$

By (empirical) Bernstein inequality plus union bound, it is easy to see $\mathbb{P}(\mathcal{E}_1 \cap \mathcal{E}_2 \cap \mathcal{E}_3) \geq 1 - \delta$.

**First we show the correctness.**

For any $t$ that $\log(t) = \mathbb{N}$, assume that $h^* \in V_t$, then we have

$$\hat{L}_t(h^*) - \hat{L}_t(\hat{h}_t) = \hat{R}_t(h^*) - \hat{R}_t(\hat{h}_t)$$

$$\leq \bar{R}_t(h^*) - \bar{R}_t(\hat{h}_t) + \sqrt{\frac{\beta_t \hat{\rho}_t(h^*, \hat{h}_t)}{t}} + \frac{\beta_t}{2t}$$

$$\leq R^* - R_*(\hat{h}_t) + \sqrt{\frac{2\beta_t \hat{\rho}_t(h^*, \hat{h}_t)}{t}} + \frac{\beta_t}{t} + \rho_*(h^*, \hat{h}_t)\frac{2C_t}{t}$$

$$\leq \sqrt{\frac{2\beta_t \hat{\rho}_t(h^*, \hat{h}_t)}{t}} + \frac{\beta_t}{t} + \rho_*(h^*, \hat{h}_t)\frac{2C_t}{t}$$

$$\leq \sqrt{\frac{2\beta_t \hat{\rho}_t(h^*, \hat{h}_t)}{t}} + \frac{\beta_t}{t} + \left( \hat{\rho}_t(h^*, \hat{h}_t) + \sqrt{\frac{2\beta_t \hat{\rho}_t(h^*, \hat{h}_t)}{t}} + \frac{\beta_t}{t} \right) \frac{2C_t}{t}$$

$$\leq \sqrt{\frac{2\beta_t \hat{\rho}_t(h^*, \hat{h}_t)}{t}} + \frac{3\beta_t}{2t} + \frac{1}{2}\hat{\rho}_t(h^*, \hat{h}_t)$$

where the first and forth inequality comes from the event $\mathcal{E}_1$ and $\mathcal{E}_3$, the second inequality comes from Lemma A.2, the third inequality comes from the definition of $R^*$ and last inequality comes from $\sqrt{\frac{2\beta_t \hat{\rho}_t(h^*, \hat{h}_t)}{t}} \leq \frac{\hat{\rho}_t(h^*, \hat{h}_t)}{2} + \frac{\beta_t}{t}$ and the assumption that $\frac{C_t}{t} \leq \frac{1}{8}$.

According to the elimination condition 10 in Algo. 1, this implies that $h^* \in V_{t+1}$. Therefore, by induction, we get that $h^* \in V_n$. By again using Lemma C.1, we can guarantee that

$$R_*(h_{out}) - R^* \leq \frac{22 \log(|\mathcal{H}|/\delta)}{n} + \frac{4R^* C_{\text{total}}}{n} + \sqrt{R^* \frac{8 \log(|\mathcal{H}|/\delta)}{n}}$$

**Next we show the sample complexity.** For any $t$ that $\log(t) = \mathbb{N}$ and any $h \in V_t$, we have

$$\Delta_h = \left( \Delta_h - (\bar{R}_t(h) - \bar{R}_t(h^*)) \right) + \left( (\bar{R}_t(h) - \bar{R}_t(h^*)) - (\hat{R}_t(h) - \hat{R}_t(h^*)) \right) + (\hat{R}_t(h) - \hat{R}_t(h^*))$$

$$\leq \frac{2C_t}{t}\rho_*(h, h^*) + \sqrt{\frac{2\beta_t \rho_*(h, h^*)}{t}} + \frac{\beta_t}{t} + \hat{R}_t(h) - \hat{R}_t(\hat{h}_t)$$

$$\leq \frac{1}{4}\rho_*(h, h^*) + \sqrt{\frac{2\beta_t \rho_*(h, h^*)}{t}} + \frac{\beta_t}{t} + \sqrt{\frac{2\beta_t \hat{\rho}_t(h^*, \hat{h}_t)}{t}} + \frac{3\beta_t}{2t} + \frac{1}{2}\hat{\rho}_t(h^*, \hat{h}_t)$$

$$\leq \frac{19}{24}\rho_*(h, h^*) + \sqrt{\frac{2\beta_t \rho_*(h, h^*)}{t}} + \sqrt{\frac{2\beta_t \hat{\rho}_t(h, h^*)}{t}} + \sqrt{\frac{2\beta_t \hat{\rho}_t(\hat{h}_t, h^*)}{t}} + \frac{6\beta_t}{t}$$

$$\leq \left( \frac{19}{24} + \frac{25}{24\beta_4} \right)\rho_*(h, h^*) + \frac{13}{24\beta_4}\rho_*(\hat{h}_t, h^*) + \left( 2\beta_4 + 6 + \frac{21}{2\beta_4} \right)$$

$$\leq \left( \frac{19}{24} + \frac{25}{24\beta_4} \right)\Delta_h + \frac{13}{24\beta_4}\Delta_{\hat{h}_t} + \left( 2\beta_4 + 6 + \frac{21}{2\beta_4} \right)\frac{\beta_t}{t} + 2 \left( \frac{19}{24} + \frac{25}{24\beta_4} + \frac{13}{24\beta_4} \right)R^*$$

$$\leq \left( \frac{19}{24} + \frac{25}{24\beta_4} \right)\Delta_h + \left( 2\beta_4 + 6 + \frac{169}{12\beta_4} + \frac{21}{2\beta_4} \right)\frac{\beta_t}{t} + 2 \left( \frac{19}{24} + \frac{25}{24\beta_4} + \frac{13}{24\beta_4} + \frac{13}{48\beta_4} \right)R^*$$

where the first inequality comes from the event $\mathcal{E}_2$ and the definition of $\hat{h}_t$, the second inequality comes from the elimination condition 10 in Algo. 1. For the third and forth inequality, we use the fact $\sqrt{AB} \leq \frac{A+B}{2}$ multiple times and the last inequality comes from Lemma C.1.

Finally, choose $\beta_4 = 25$ and solve this inequality, we get $\Delta_h \leq \frac{120\beta_t}{t} + 12R^*$

Therefore, we get the probability of query as

$$\mathbb{P}\left(x_{t+1} \in \mathrm{Dis}(V_{t+1})\right) \leq \mathbb{P}\left(\exists h \in V_{t+1} : h(x_t) \neq h^*(x_t), \Delta_h \leq \frac{120\beta_t}{t} + 12R^*\right)$$

$$\leq \mathbb{P}\left(\exists h \in V_{t+1} : h(x_t) \neq h^*(x_t), \rho_*(h, h^*) \leq 14R^* + \frac{120\beta_t}{t}\right)$$

$$\leq \theta^*(14R^* + \frac{120\beta_t}{t})\left(14R^* + \frac{120\beta_t}{t}\right)$$

Therefore, we get the final prove by summing this probability over all the time.

### C.2 Why vanilla Robust CAL does not work?

**Proposition C.1.** *When $R^* \gg 0$ and the corruptions are unknown to the learner, there exists an instance and an adversary such that the vanilla Robust CAL can never output the target hypothesis.*

*Proof.* Suppose $\mathcal{X} = \{x_1, x_2, x_3\}$ where $\nu_*(x_1) = \xi_1 \gg 0, \nu_*(x_2) = \xi_2 \leq \frac{\xi_1}{64}$ and $\nu_*(x_3) = 1 - \xi_1 - \xi_2$. Here we further assume that $\nu$ is given to learner. For labels, we set $\eta_*^{x_1} = \frac{1}{2}, \eta_*^{x_2} = \eta_*^{x_2} = 1$. Now consider $h_1 : h_1(x_1) = h_1(x_2) = h_1(x_3) = 1$ and $h_2 : h_2(x_1) = h_2(x_2) = 0, h_2(x_3) = 1$. With some routine calculations, we can obtain that:

$$R^* = R_*(h_1) = \frac{1}{2}\xi_1, \quad R_*(h_2) = \frac{1}{2}\xi_1 + \xi_2, \quad \rho_*(h_1, h_2) = \xi_1 + \xi_2$$

Now suppose the adversary corrupts $\eta_*^{x_1}$ from $\frac{1}{2}$ to $\eta_s^{x_1} = \frac{15}{32}$ for all $s \leq \tau$ and will stop corrupting at certain time $\tau$. Consider this case $C_t \leq \frac{1}{32}t$, which satisfies our corruption assumption.

With such corruptions, we have that for any $t \leq \tau$,

$$\bar{R}_t(h_1) = \frac{17}{32}\xi_1, \quad \bar{R}_t(h_2) = \frac{15}{32}\xi_1 + \xi_2,$$

Since $\bar{R}_t(h_2) \geq \bar{R}_t(h_1)$, so $h_2$ will never be eliminated before $\tau$. Next we show that $h_1$ can be eliminated before $\tau$. Note that, when $\tau \geq O(\frac{1}{\xi_1})$, we can always find a proper $t \leq \tau$ such that

$$\hat{R}_t(h_1) - \hat{R}_t(h_2) \geq \frac{1}{16}\xi_1 - \xi_2 - \widetilde{\mathcal{O}}\left(\sqrt{\frac{\xi_1 + \xi_2}{t}} + \frac{1}{t}\right)$$

In the non-corrupted setting, the confidence threshold of vanilla Robust CAL is always $\widetilde{\mathcal{O}}\left(\sqrt{\frac{\xi_1+\xi_2}{t}} + \frac{1}{t}\right)$, which can be smaller than $\frac{1}{16}\xi_1 - \xi_2 - \widetilde{\mathcal{O}}\left(\sqrt{\frac{\xi_1+\xi_2}{t}} + \frac{1}{t}\right)$ for large enough $t$, so the above inequality shows that $h_1$ can be eliminated before $\tau$. This implies that, if our target accuracy $\varepsilon < \xi_2$, then the vanilla Robust CAL will never able to output the correct answer no matter how many unlabeled samples are given. On the other hand, in the passive learning, one can still output the target $h_1$ as long as $n \gg \tau$. □

## D More detailed explanation for CALRuption for line 9 to 13

Here we provide a more detailed explanation on line 9 13

- In Line 9, we are going to estimate the underlying distribution of samples based on the collected samples. To be specific, we have the estimated gap between each pair of $h$ and $h'$, so the initial desire is to find a proper distribution that induces all gaps uniformly close to all the estimated gaps. But this is impossible, so we instead choose the distribution that minimizes the worst-case pairs scaled with its variance. With such an estimated distribution, we can naturally get the estimated error of each hypothesis $h$ denoted as $R_{\hat{\mathcal{D}}}(h)$.

- In Line 10, recall that we already have the $R_{\hat{\mathcal{D}}}(h)$, and the previously estimated gap between any hypothesis $h$ and the previous estimated best hypothesis $\hat{h}_*^{l-1}$, denoted as $\hat{\Delta}_h^{l-1}$.

So based on these two terms, we can have a pessimistic estimation of the current best hypothesis $\hat{h}_*^{l-1}$.

- Then in Line 11, based on the estimated best hypothesis $\hat{h}_*^{l-1}$, we can further have a new estimated gap $\hat{\Delta}_h^l$.

Up to this point, we have an estimate of the performance of each hypothesis ( $\hat{\Delta}_h^l$ ). Now recall that in the traditional elimination-style algorithms like Robust CAL, we will permanently eliminate all the hypotheses for which $\hat{\Delta}_h^l$ is larger than some threshold and then do a disagreement-based query on the remaining hypothesis set. But here, the learner never makes a "hard" decision to eliminate any hypothesis. Instead, it assigns different query probability to each based on the estimated gap $\hat{\Delta}_h^l$ for each hypothesis, That is what Line 12 and Line 13 are doing. To be specific:

- In Line 12, we divide the hypothesis into $l+1$ sets based on $\hat{\Delta}_h^l$. Again in the traditional elimination-style algorithm, the only remaining active hypothesis set is $V_{l+1}^l$.
- In Line 13, based on these layered hypothesis sets, we are going to assign the query probability on the incoming $x$. Intuitively, for each $x$, we want to find the lowest policy set it belongs to, among all those layered sets. Then, because the lower the set is, the smaller its corresponding estimated gap is, so intuitively, we want to assign a higher query probability to those that have a lower corresponding hypothesis set.

# E    Analysis for CALRuption

## E.1    Notations

Let $\mathcal{I}_l$ denotes the epoch $l$, $C_l$ denotes $C_{\mathcal{I}_l}$.

## E.2    Concentration guarantees on $\delta$-robust estimator

In this section, we show the analysis by using the Catoni's estimator which is described in detail as below. Note that the same estimator has been used in previous works including Wei et al. [2020], Camilleri et al. [2021], Lee et al. [2021].

**Lemma E.1.** *(Concentration inequality for Catoni's estimator Wei et al. [2020]) Let $\mathcal{F}_0 \subset \cdots \subset \mathcal{F}_n$ be a filtration, and $X_1, \ldots, X_n$ be real random variables such that $X_i$ is $\mathcal{F}_i$ -measurable, $\mathbb{E}[X_i \mid \mathcal{F}_{i-1}] = \mu_i$ for some fixed $\mu_i$, and $\sum_{i=1}^n \mathbb{E}\left[(X_i - \mu_i)^2 \mid \mathcal{F}_{i-1}\right] \leq V$ for some fixed V. Denote $\mu \triangleq \frac{1}{n} \sum_{i=1}^n \mu_i$ and let $\widehat{\mu}_{n,\alpha}$ be the Catoni's robust mean estimator of $X_1, \ldots, X_n$ with a fixed parameter $\alpha > 0$, that is, $\widehat{\mu}_{n,\alpha}$ is the unique root of the function*

$$f(z) = \sum_{i=1}^n \psi\left(\alpha\left(X_i - z\right)\right)$$

*where*

$$\psi(y) = \begin{cases} \ln\left(1 + y + y^2/2\right), & \text{if } y \geq 0 \\ -\ln\left(1 - y + y^2/2\right), & \text{else} \end{cases}$$

*Then for any $\delta \in (0, 1)$, as long as $n$ is large enough such that $n \geq \alpha^2\left(V + \sum_{i=1}^n (\mu_i - \mu)^2\right) + 2\log(1/\delta)$, we have with probability at least $1 - 2\delta$,*

$$|\widehat{\mu}_{n,\alpha} - \mu| \leq \frac{\alpha\left(V + \sum_{i=1}^n (\mu_i - \mu)^2\right)}{n} + \frac{2\log(1/\delta)}{\alpha n}$$

$$\leq \frac{\alpha\left(V + \sum_{i=1}^n \mu_i^2\right)}{n} + \frac{2\log(1/\delta)}{\alpha n}.$$

**Lemma E.2** (Concentration inequality in our case). *For any fixed epoch $l$ and any pair of classifier $h, h' \in \mathcal{H}$, as long as $N_l \geq 4\log(1/\delta)$, with probability at least $1 - \delta$, we have*

$$|(\hat{R}_l(h) - \hat{R}_l(h')) - W_l^{h,h'}| \leq \sqrt{\frac{10\log(1/\delta)\hat{\rho}_l(h, h')}{N_l \min_{x \in Dis(h,h')} q_l^x}}$$

*where $\hat{R}_l(h) = \frac{1}{|\mathcal{I}_l|} \sum_{t \in \mathcal{I}} \mathbb{E}_{y \sim \text{Ber}(\eta_t^{x_t})} [\mathbf{1}\{h(x_t) \neq y\}]$ (restate)*

*Proof.* First we calculate the expectation and variance of $(\hat{\ell}_t(h) - \hat{\ell}_t(h'))$ for each $t \in \mathcal{I}_l$,

$$\mathbb{E}_{y \sim \text{Ber}(\eta_t^{x_t})} \mathbb{E}_{Q_t} \left[ \hat{\ell}_t(h) - \hat{\ell}_t(h') \right] = \mathbb{E}_{y \sim \text{Ber}(\eta_t^{x_t})} \left[ \mathbf{1}\{h(x_t) \neq y\} - \mathbf{1}\{h'(x_t) \neq y\} \right]$$
$$\leq \mathbf{1}\{h(x_t) \neq h'(x_t)\}$$

and,

$$\text{Var}_t \left( \hat{\ell}_t(h) - \hat{\ell}_t(h') \right) \leq \mathbb{E}_{y \sim \text{Ber}(\eta_t^{x_t})} \mathbb{E}_{Q_t} \left[ \left( \hat{\ell}_t(h) - \hat{\ell}_t(h') \right)^2 \right]$$
$$= \mathbb{E}_{y \sim \text{Ber}(\eta_t^{x_t})} \mathbb{E}_{Q_t} \left[ \frac{\mathbf{1}\{h(x_t) \neq h'(x_t)\}}{(q_l^{x_t})^2} \right]$$
$$= \frac{\mathbf{1}\{h(x_t) \neq h'(x_t)\}}{q_l^{x_t}}$$
$$\leq \frac{\mathbf{1}\{h(x_t) \neq h'(x_t)\}}{\min_{x' \in \text{Dis}(h,h')} q_l^{x'}}$$

Then according to the Lemma E.1, we have

$$|(\hat{R}_l(h) - \hat{R}_l(h')) - W_l^{h,h'}|$$
$$\leq \frac{\alpha_l^{h,h'} \left( \frac{\sum_t \mathbf{1}\{h(x_t) \neq h'(x_t)\}}{\min_{x' \in \text{Dis}(h,h')} q_l^{x'}} + \sum_t \mathbf{1}\{h(x_t) \neq h'(x_t)\} \right)}{N_l} + \frac{2 \log(1/\delta)}{\alpha_l^{h,h'} N_l}$$
$$\leq \frac{2\alpha_l^{h,h'} \hat{\rho}_l(h, h')}{\min_{x' \in \text{Dis}(h,h')} q_l^{x'}} + \frac{2 \log(1/\delta)}{\alpha_l^{h,h'} N_l}$$
$$= \sqrt{\frac{10 \log(1/\delta) \hat{\rho}_l(h, h')}{N_l \min_{x \in \text{Dis}(h,h')} q_l^x}}$$

The last one comes from choosing $\alpha_l^{h,h'} = \sqrt{\frac{2 \log(1/\delta) \min_{x \in \text{Dis}(h,h')} q_l^x}{5 N_l \hat{\rho}_l(h,h')}}$ and also it is easy to verify that

$$(\alpha_l^{h,h'})^2 \left( \frac{N_l \hat{\rho}_l(h, h')}{\min_{x' \in \text{Dis}(h,h')} q_l^{x'}} + \sum_t ((R_*(h) - R_*(h')) - (R_t(h) - R_t(h')))^2 \right) + 2 \log(1/\delta)$$
$$\leq 4 \log(1/\delta) \leq N_l.$$

$\square$

### E.3 High probability events

Define the event $\mathcal{E}_{gap}$ as

$$\mathcal{E}_{gap} := \left\{ \forall l, \forall h, h' \in \mathcal{H}, |(\hat{R}_l(h) - \hat{R}_l(h')) - W_l^{h,h'}| \leq \sqrt{\frac{10 \beta_3 \hat{\rho}_l(h, h')}{N_l \min_{x \in \text{Dis}(h,h')} q_l^x}} \right\},$$

and event $\mathcal{E}_{dis1}, \mathcal{E}_{dis2}$ as

$$\mathcal{E}_{dis1} := \left\{ \forall l, \forall h, h' \in \mathcal{H}, |\hat{\rho}_l(h, h') - \rho_*(h, h')| \leq \sqrt{\frac{\beta_3 \hat{\rho}_l(h, h')}{N_l}} + \frac{\beta_3}{N_l} \right\}$$

$$\mathcal{E}_{dis2} := \left\{ \forall l, \forall h, h' \in \mathcal{H}, |\hat{\rho}_l(h, h') - \rho_*(h, h')| \leq \sqrt{\frac{\beta_3 \rho_*(h, h')}{N_l}} + \frac{\beta_3}{N_l} \right\}.$$

By condition 1 of $\delta$-robust estimator in Algo 2, the (empirical) Bernstein inequality and the union bounds, we have easily get $\mathbb{P}(\mathcal{E}_{gap} \cap \mathcal{E}_{dis1} \cap \mathcal{E}_{dis2}) \geq 1 - \delta$ as shown in the following lemmas.

**Lemma E.3.** $\mathbb{P}(\mathcal{E}_{est}) \geq 1 - \delta/3$

*Proof.* We prove this by condition 1 in Algo 2 and the union bound over $|\mathcal{H}|^2$ number of hypothesis pairs and $\frac{1}{2}\lfloor \log(n) \rfloor$ number of epochs. □

**Lemma E.4.** $\mathbb{P}(\mathcal{E}_{gap1}) \geq 1 - \delta/3, \mathbb{P}(\mathcal{E}_{gap2}) \geq 1 - \delta/3$

*Proof.* We prove this by (empirical) Bernstein inequality in Algo 2 and the union bound over $|\mathcal{H}|^2$ number of hypothesis pairs and $\frac{1}{2}\lfloor \log(n) \rfloor$ number of epochs. □

### E.4 Gap estimation accuracy

In this section, we show that $\hat{\Delta}_h^l$ is close to $\Delta_h$ for all $l, h$. To prove this, we first show some auxiliary lemmas as follows.

**Lemma E.5** (Estimation accuracy for $\hat{\mathcal{D}}_l$). *On event $\mathcal{E}_{gap}$, for any fixed epoch $l$, for any fixed pair $h, h' \in \mathcal{H}$, suppose $j = \max\{i | h, h' \in V_l^i\}$, we have*

$$|(R_{\hat{\mathcal{D}}_l}(h) - R_{\hat{\mathcal{D}}_l}(h')) - (R_*(h) - R_*(h'))|$$
$$\leq \frac{1}{16}\left(\max\{\hat{\Delta}_h^{l-1}, \hat{\Delta}_{h'}^{l-1}\} + \epsilon_l\right) + \frac{4C_l}{N_l}R^* + \frac{2C_l}{N_l}\max\{\Delta_h, \Delta_{h'}\}$$

*Proof.* Firstly we show that, for any pair $h, h' \in \mathcal{H}$, we have

$$|(R_{\hat{\mathcal{D}}_l}(h) - R_{\hat{\mathcal{D}}_l}(h')) - (R_*(h) - R_*(h'))|$$
$$\leq |(R_{\hat{\mathcal{D}}_l}(h) - R_{\hat{\mathcal{D}}_l}(h')) - W_l^{h,h'}| + |W_l^{h,h'} - (\hat{R}_l(h) - \hat{R}_l(h'))| + |(\hat{R}_l(h) - \hat{R}_l(h')) - (R_*(h) - R_*(h'))|$$
$$\leq \max_{h_1, h_2 \in \mathcal{H}} \left|\left((R_{\hat{\mathcal{D}}_l}(h_1) - R_{\hat{\mathcal{D}}_l}(h_2)) - W_l^{h_1,h_2}|\sqrt{\frac{\min_{x \in \text{Dis}(h_1,h_2)} q_l^x}{\hat{\rho}_l(h_1, h_2)}}\right)\sqrt{\frac{\hat{\rho}_l(h, h')}{\min_{x \in \text{Dis}(h,h')} q_l^x}}\right.$$
$$+ |W_l^v - (\hat{R}_l(h) - \hat{R}_l(h'))| + |(\hat{R}_l(h) - \hat{R}_l(h')) - (R_*(h) - R_*(h'))|$$
$$\leq \max_{h_1, h_2 \in \mathcal{H}} \left|\left((\hat{R}_l(h_1) - \hat{R}_l(h_2)) - W_l^{h_1,h_2}|\sqrt{\frac{\min_{x \in \text{Dis}(h_1,h_2)} q_l^x}{\hat{\rho}_l(h_1, h_2)}}\right)\sqrt{\frac{\hat{\rho}_l(h, h')}{\min_{x \in \text{Dis}(h,h')} q_l^x}}\right.$$
$$+ |W_l^v - (\hat{R}_l(h) - \hat{R}_l(h'))| + |(\hat{R}_l(h) - \hat{R}_l(h')) - (R_*(h) - R_*(h'))|$$
$$\leq 2 \max_{h_1, h_2 \in \mathcal{H}} \left|\left((\hat{R}_l(h_1) - \hat{R}_l(h_2)) - W_l^{h_1,h_2}|\sqrt{\frac{\min_{x \in \text{Dis}(h_1,h_2)} q_l^x}{\hat{\rho}_l(h_1, h_2)}}\right)\sqrt{\frac{\hat{\rho}_l(h, h')}{\min_{x \in \text{Dis}(h,h')} q_l^x}}\right.$$
$$+ |(\hat{R}_l(h) - \hat{R}_l(h')) - (R_*(h) - R_*(h'))|$$
$$\leq 2\sqrt{\frac{10\beta_3}{N_l}}\sqrt{\frac{\hat{\rho}_l(h, h')}{\min_{x \in \text{Dis}(h,h')} q_l^x}} + |(\hat{R}_l(h) - \hat{R}_l(h')) - (R_*(h) - R_*(h'))|$$

The third inequality comes from the definition of $\hat{\mathcal{D}}_l$ and the last inequality comes from the Condition 1 of $\delta$-robust estimator in Algo. 2.

For the first term, for any $x \in \text{Dis}(h, h')$, by the definition of $q_l^x$ in line 13 and the fact that $(h, h') \in \mathcal{Z}(x)$, we have that,

$$q_l^x \geq \frac{\beta_1 \hat{\rho}_l(h, h')}{N_l}\epsilon_j^{-2}. \quad , \text{where } j = \max\{i \in [l-1] \mid h, h' \in V_l^i\}$$

So we can further lower bound the $\min_{x \in \text{Dis}(h,h')} q_l^x$ by

$$\min_{x \in \text{Dis}(h,h')} q_l^x \geq \frac{\beta_1 \hat{\rho}_l(h, h')}{N_l}\epsilon_j^{-2} \quad , \text{where } j = \max\{i \in [l-1] \mid h, h' \in V_l^i\}$$

and therefore upper bound the first term as

$$2\sqrt{\frac{10\beta_3}{N_l}}\sqrt{\frac{\hat{\rho}_l(h, h')}{\min_{x \in \text{Dis}(h,h')} q_l^x}} \leq 2\sqrt{\frac{10\beta_3}{\beta_1}}\epsilon_j.$$

For the second term, by the definition of corruptions, we have

$$|(\hat{R}_l(h) - \hat{R}_l(h')) - (R_*(h) - R_*(h'))|$$

$$\leq |(\hat{R}_l(h) - \hat{R}_l(h')) - (\overline{R}_l(h) - \overline{R}_l(h'))| + |(\overline{R}_l(h) - \overline{R}_l(h')) - (R_*(h) - R_*(h'))|$$

$$\leq 2\sqrt{\frac{\beta_3}{N_l}} + \frac{2C_l}{N_l}\rho_*(h, h')$$

$$\leq 2\sqrt{\frac{\beta_3}{\beta_1}\epsilon_l} + \frac{2C_l}{N_l}\left(\rho_*(h, h^*) + \rho_*(h', h^*)\right)$$

$$\leq 2\sqrt{\frac{\beta_3}{\beta_1}\epsilon_l} + \frac{4C_l}{N_l}R^* + \frac{2C_l}{N_l}\max\{\Delta_h, \Delta_{h'}\}$$

where the second inequality comes from Bernstein inequality and Lemma A.2.

Finally we are going to make the connection between $\epsilon_j$ and the $\hat{\Delta}_h^{l-1}, \hat{\Delta}_{h'}^{l-1}$. Note that if $j < l - 1$, by definition of $j$, we must have $h, h' \notin V_l^{j+1}$. By the definition that $\forall h \notin V_{l+1}^i, \hat{\Delta}_h^i \geq \epsilon_i$, we have

$$\max\{\hat{\Delta}_h^{l-1}, \hat{\Delta}_{h'}^{l-1}\} > \epsilon_{j+1} = \frac{\epsilon_j}{2}.$$

and if $j = l - 1$, we directly have $\frac{\epsilon_j}{2} \leq \epsilon_l$. Therefore, we have $\frac{\epsilon_j}{2} \leq \max\{\hat{\Delta}_h^{l-1}, \hat{\Delta}_{z'}^{l-1}\} + \epsilon_l$. $\quad\square$

**Lemma E.6** (Upper bound of the estimated gap). *On event $\mathcal{E}_{gap}$, for any fixed epoch $l$, suppose its previous epoch satisfies that, for all $h \in \mathcal{H}$,*

$$\Delta_h \leq \frac{3}{2}\hat{\Delta}_h^{l-1} + \frac{3}{2}\epsilon_{l-1} + 3g_{l-1}, \tag{2}$$

$$\hat{\Delta}_h^{l-1} \leq 2\left(\Delta_h + \epsilon_{l-1} + g_{l-1}\right), \tag{3}$$

*then we have,*

$$\hat{\Delta}_h^l \leq 2\left(\Delta_h + \epsilon_l + g_l\right)$$

*where $g_l = \frac{2}{\beta_1}\epsilon_l^2 \sum_{s=1}^l C_s \left(2R^*\mathbf{1}\left\{\frac{2C_{\mathcal{I}_s}}{N_s} \leq \frac{1}{16}\right\} + \mathbf{1}\left\{\frac{2C_{\mathcal{I}_s}}{N_s} > \frac{1}{16}\right\}\right).$*

*Proof.* According to the definition of $\hat{\Delta}_h^l$, If $\left\langle h - \hat{h}_*^l, \hat{\theta}_l \right\rangle - \beta_2 \hat{\Delta}_{\hat{h}_*^l}^{l-1} \leq \epsilon_l$, then the above trivially holds, Otherwise, we have

$$\hat{\Delta}_h^l = R_{\hat{\mathcal{D}}_l}(h) - \left(R_{\hat{\mathcal{D}}_l}(\hat{h}_*^l) + \beta_2 \hat{\Delta}_{\hat{h}_*^l}^{l-1}\right)$$

$$= \left((R_{\hat{\mathcal{D}}_l}(h) - R_{\hat{\mathcal{D}}_l}(\hat{h}_*^l)) - (R_*(h) - R_*(\hat{h}_*^l))\right) + (R_*(h) - R_*(\hat{h}_*^l)) - \beta_2 \hat{\Delta}_{\hat{h}_*^l}^{l-1}$$

$$\leq \left((R_{\hat{\mathcal{D}}_l}(h) - R_{\hat{\mathcal{D}}_l}(\hat{h}_*^l)) - (R_*(h) - R_*(\hat{h}_*^l))\right) + \Delta_h - \beta_2 \hat{\Delta}_{\hat{h}_*^l}^{l-1}$$

$$\leq \frac{1}{16}\left(\max\{\hat{\Delta}_h^{l-1}, \hat{\Delta}_{\hat{h}_*^l}^{l-1}\} + \epsilon_l\right) + \frac{1}{16}\max\{\Delta_h, \Delta_{\hat{h}_*^l}\} + \Delta_h - \beta_2 \hat{\Delta}_{\hat{h}_*^l}^{l-1}$$

$$+ \underbrace{\frac{4C_l}{N_l}R^*\mathbf{1}\{\frac{2C_l}{N_l} \leq \frac{1}{16}\} + \frac{2C_l}{N_l}\mathbf{1}\{\frac{2C_l}{N_l} > \frac{1}{16}\}}_{\text{Corruption Term}}$$

$$= \frac{1}{16}(\hat{\Delta}_h^{l-1} + \epsilon_l) + \frac{1}{16}\Delta_h + \frac{1}{16}\hat{\Delta}_{\hat{h}_*^l}^{l-1} + \frac{1}{16}\Delta_{\hat{h}_*^l} - \beta_2 \hat{\Delta}_{\hat{h}_*^l}^{l-1} + \Delta_h + \text{Corruption Term}$$

$$\leq \left(\frac{1}{16}(\hat{\Delta}_h^{l-1} + \epsilon_l) + \frac{1}{16}\Delta_h + \Delta_h\right) + \left(\frac{1}{16}\hat{\Delta}_{\hat{h}_*^l} + \frac{3}{32}\hat{\Delta}_{\hat{h}_*^l}^{l-1} - \beta_2 \hat{\Delta}_{\hat{h}_*^l}^{l-1}\right) + \frac{3}{32}(\epsilon_{l-1} + 2g_{l-1}) + \text{Corruption Term}$$

$$\leq \left(\frac{1}{16}(\hat{\Delta}_h^{l-1} + \epsilon_l) + \frac{1}{16}\Delta_h + \Delta_h\right) + \frac{3}{32}(\epsilon_{l-1} + 2g_{l-1}) + \text{Corruption Term}$$

$$= \frac{1}{16}\hat{\Delta}_h^{l-1} + \left(1 + \frac{1}{16}\right)\Delta_h + \frac{1}{4}\epsilon_l + 4R^*\frac{C_l}{N_l} + \frac{3}{16}g_{l-1}$$

$$\leq 2(\Delta_h + \epsilon_l + g_l)$$

Here the first inequality comes from the definition of $h^*$, the second inequality comes from Lemma E.5, the third inequality comes from the the assumption (1) and the penultimate inequality comes from the fact that $\beta_2 \geq \frac{5}{32}$. Finally, the last inequality comes from assumption (2).

$\square$

**Lemma E.7** (Lower bound of the estimated gap)**.** *On event $\mathcal{E}_{gap}$, for any fixed epoch $l$, suppose the following holds, for all $h \in \mathcal{H}$,*

$$\hat{\Delta}_h^{l-1} \leq 2\left(\Delta_h + \epsilon_{l-1} + g_{l-1}\right), \tag{4}$$

*then we have,*

$$\Delta_h \leq \frac{3}{2}\hat{\Delta}_h^l + \frac{3}{2}\epsilon_l + 3g_l$$

*Proof.*

$$
\begin{aligned}
\hat{\Delta}_h^l &\geq R_{\hat{\mathcal{D}}_l}(h) - \left(R_{\hat{\mathcal{D}}_l}(h^*) + \beta_2 \hat{\Delta}_{h^*}\right) \\
&= \left(\left(R_{\hat{\mathcal{D}}_l}(h) - R_{\hat{\mathcal{D}}_l}(h^*)\right) - \left(R_*(h) - R^*\right)\right) + \Delta_h - \beta_2 \hat{\Delta}_{h^*}^{l-1} \\
&\geq -\frac{1}{16}(\hat{\Delta}_h^{l-1} + \epsilon_l) - \frac{1}{16}\Delta_h - \frac{1}{16}\hat{\Delta}_{h^*}^{l-1} - \frac{1}{16}\Delta_{h^*} - \beta_2 \hat{\Delta}_{h^*}^{l-1} + \Delta_h \\
&\quad - \underbrace{\left(4R^* \frac{C_l}{N_l} \mathbf{1}\left\{\frac{2C_l}{N_l} \leq \frac{1}{16}\right\} + \frac{C_l}{N_l} \mathbf{1}\left\{\frac{2C_l}{N_l} > \frac{1}{16}\right\}\right)}_{\text{Corruption Term}} \\
&= -\frac{1}{16}(\hat{\Delta}_h^{l-1} + \epsilon_l) - \frac{1}{16}\Delta_h - \frac{1}{16}\hat{\Delta}_{h^*} - \beta_2 \hat{\Delta}_{h^*}^{l-1} + \Delta_h - \text{Corruption Term} \\
&\geq -\frac{1}{16}(2\Delta_h + 2\epsilon_{l-1} + 2g_{l-1} + \epsilon_l) + \Delta_h - (\frac{1}{16} + \beta_2)(2\epsilon_{l-1} + 2g_{l-1}) - \text{Corruption Term} \\
&\geq \frac{13}{16}\Delta_h - \frac{38}{32}\epsilon_l - \frac{18}{32}g_{l-1} - 4R^* \frac{C_l}{N_l} - \text{Corruption Term} \\
&\geq \frac{13}{16}\Delta_h - \frac{38}{32}\epsilon_l - \frac{18}{8}g_l
\end{aligned}
$$

Here the first inequality comes from the definition of $\hat{h}_*^l$, the second inequality comes from Lemma E.5. and the third inequality comes from the upper bound of the estimated gap in Lemma E.6. $\square$

**Now we are ready to prove the final key lemma, which shows that such upper bound and lower bound for $\hat{\Delta}_h^l$ holds for all $l$ and $h$.**

**Lemma E.8** (Upper bound and lower bound for all estimation)**.** *On event $\mathcal{E}_{gap}$, for any epoch $l$, for all $h \in \mathcal{H}$,*

$$\hat{\Delta}_h^l \leq 2\left(\Delta_h + \epsilon_l + g_l\right) \tag{5}$$

$$\Delta_h \leq \frac{3}{2}\hat{\Delta}_h^l + \frac{3}{2}\epsilon_l + 3g_l \tag{6}$$

*Proof.* We prove this by induction.

For the base case where $l = 1$. we can easily have the following

$$\hat{\Delta}_h^1 \leq 1 \leq 2\Delta_h + 2\epsilon_1 + 2g_l$$

and also, by using Lemma E.7 and the fact that $\hat{\Delta}_h^0 \leq 2(\Delta_h + \epsilon_0 + g_0)$, it is easy to get

$$\Delta_h \leq \frac{3}{2}\hat{\Delta}_h^1 + \frac{3}{2}\epsilon_1 + 3g_1$$

So the target inequality holds for $l = 1$.

Suppose the target inequality holds for $l' - 1$ where $l' \geq 2$, then by Lemma E.6, we show that the first target inequality holds for $l'$. Also by Lemma E.7, we show that the second target inequality holds for $l'$. Therefore, we finish the proof.

$\square$

## E.5 Auxiliary lemmas

**Lemma E.9.** *For any epoch $l$ and layer $j$, we have*

$$\max_{h \in V_l^j} \rho_*(h, h^*) \leq 2R^* + 3\epsilon_j + 3g_{l-1}$$

*Proof.*

$$\max_{h \in V_l^j} \rho_*(h, h^*) \leq 2R^* + \max_{h \in V_l^j} \Delta_h$$

$$\leq 2R^* + \max_{h \in V_l^j} \left( \frac{3}{2} \hat{\Delta}_h^{l-1} + \frac{3}{2} \epsilon_{l-1} + 3g_{l-1} \right)$$

$$\leq 2R^* + 3\epsilon_j + 3g_{l-1}$$

The first inequality comes from the fact the $\rho_*(h, h^*) \leq R_*(h) + R^* = 2R^* + \Delta_h$, the second inequality comes form the lower bound in Lemma E.8 and the last inequality is by the definition of $V_l^j$. $\qquad\square$

## E.6 Main proof for Theorem 5.1

Here we assume $\log_4(\frac{n}{\beta_1}) \notin \mathbb{N}$ and there are no corruptions in the last unfinished epoch $\lceil \log_4(\frac{n}{\beta_1}) \rceil$. This will not effect the result but will make the proof easier. Given that events $\mathcal{E}_{gap}, \mathcal{E}_{dis1}$ and $\mathcal{E}_{dis2}$, then we have the following proofs.

**First we deal with the sample complexity.**

For any $t \in \mathcal{I}_l$, the probability of $x_t$ being queried $(Q_t)$ is

$$\mathbb{E}[Q_t] = \sum_{x \in \mathcal{X}} P(x_t = x) q_l^x$$

$$= \sum_{x \in \mathcal{X}} P(x_t = x) \max_{(h,h') \in \mathcal{Z}(x)} \frac{\beta_1 \hat{\rho}_{l-1}(h, h')}{N_l} \epsilon_{k(h,h',l)}^{-2}$$

$$\leq \frac{\beta_1}{N_l} \sum_{x \in \mathcal{X}} P(x_t = x) \max_{(h,h') \in \mathcal{Z}(x)} \rho_*(h, h') \epsilon_{k(h,h',l)}^{-2}$$

$$+ 4 \frac{\beta_1}{N_l} \sum_{x \in \mathcal{X}} P(x_t = x) \sqrt{\rho_*(h, h') \epsilon_{k(h,h',l)}^{-2}} + \frac{4\beta_1}{N_l}$$

$$\leq 5 \frac{\beta_1}{N_l} \sum_{x \in \mathcal{X}} P(x_t = x) \max_{(h,h') \in \mathcal{Z}(x)} \rho_*(h, h') \epsilon_{k(h,h',l)}^{-2} + 8 \frac{\beta_1}{N_l}$$

$$= 5 \frac{\beta_1}{N_l} \sum_{x \in \mathcal{X}} P(x_t = x) \rho_*(h_1^x, h_2^x) \epsilon_{j^x}^{-2} + 8 \frac{\beta_1}{N_l}$$

$$\leq 5 \frac{\beta_1}{N_l} \sum_{x \in \mathcal{X}} P(x_t = x) \max_{h_3,h_4 \in V_l^{j^x}} \rho_*(h_3, h_4) \epsilon_{j^x}^{-2} + 8 \frac{\beta_1}{N_l}$$

$$\leq 10 \frac{\beta_1}{N_l} \sum_{x \in \mathcal{X}} P(x_t = x) \max_{h \in V_l^{j^x}} \rho_*(h, h^*) \epsilon_{j^x}^{-2} + 8 \frac{\beta_1}{N_l}$$

$$\leq 10 \frac{\beta_1}{N_l} \sum_{x \in \mathcal{X}} P(x_t = x) \left( 2R^* \epsilon_{j^x}^{-2} + 3\epsilon_{j^x}^{-1} + 3g_{l-1} \epsilon_{j^x}^{-2} \right) + 8 \frac{\beta_1}{N_l}$$

$$= 10 \frac{\beta_1}{N_l} \sum_{i=1}^{l-1} \left( 2R^* \epsilon_i^{-2} + 3\epsilon_i^{-1} + 3g_{l-1} \epsilon_i^{-2} \right) \sum_{x \in \mathcal{X}} P(x_t = x) \mathbf{1}\{j^x = i\} + 8 \frac{\beta_1}{N_l}$$

$$\leq 10 \frac{\beta_1}{N_l} \sum_{i=0}^{l-1} \left( 2R^* \epsilon_i^{-2} + 3\epsilon_i^{-1} + 3g_{l-1} \epsilon_i^{-2} \right) \mathbb{P}(x \in \text{Dis}(V_l^i)) + 8 \frac{\beta_1}{N_l}$$

Here $(h_1^x, h_2^x) = \arg\max_{(h,h')\in\mathcal{Z}(x)} \rho_*(h,h')\epsilon_{k(h,h',l)}^{-2}$ and $j^x = k(h_1^x, h_2^x, l)$. The first inequality comes from the event $\mathcal{E}_{dis2}$, the second inequality comes from the fact that $\sqrt{\rho_*(h,h')\epsilon_{k(h,h',l)}^{-2}} \leq \rho_*(h,h')\epsilon_{k(h,h',l)}^{-2} + 1$ and penultimate inequality comes from the Lemma E.9.

Now we can use the standard techniques to bound $\mathbb{P}(x \in \mathrm{Dis}(V_l^i))$ as follows

$$
\begin{aligned}
\mathbb{P}(x \in \mathrm{Dis}(V_l^i)) &= \mathbb{P}\left(\exists h, h' \in V_l^i : h(x) \neq h'(x)\right) \\
&\leq \mathbb{P}\left(\exists h \in V_l^i : h(x) \neq h^*(x)\right) \\
&\leq \mathbb{P}\left(\exists h \in \mathcal{H} : h(x) \neq h^*(x), \rho_*(h,h^*) \leq 2R^* + 3\epsilon_i + 3g_{l-1}\right) \\
&\leq \theta^*(2R^* + 3\epsilon_i + g_{l-1})(2R^* + 3\epsilon_i + 3g_{l-1})
\end{aligned}
$$

where again the first inequality comes from Lemma E.9.

Combine with the above result, we get the expected number of queries inside a complete epoch $l$ as,

$$
\begin{aligned}
\sum_{t\in\mathcal{I}_l} \mathbb{E}[Q_t] &= 10\beta_1 \sum_{i=0}^{l-1} \theta^*(2R^* + 3\epsilon_i + g_{l-1}) \\
&\quad * \left(4(R^*)^2\epsilon_i^{-2} + 12R^*\epsilon_i^{-1} + 12R^*g_{l-1}\epsilon_i^{-2} + 18g_{l-1}\epsilon_i^{-1} + 9g_{l-1}^2\epsilon_i^{-2} + 9\right) \\
&\leq 20\beta_1\theta^*(2R^* + 3\epsilon_{l-1} + g_{l-1}) \\
&\quad * \left(4(R^*)^2\epsilon_l^{-2} + 12R^*\epsilon_l^{-1} + \frac{24}{\beta_1}R^*\bar{C}_{l-1} + \frac{36}{\beta_1}\bar{C}_{l-1}\epsilon_{l-1} + \frac{36}{\beta_1^2}\bar{C}_{l-1}^2\epsilon_{l-1}^2 + 9\right) \\
&\leq 20\beta_1\theta^*(2R^* + 3\epsilon_{l-1} + g_{l-1}) * \left(4(R^*)^2\epsilon_l^{-2} + 12R^*\epsilon_l^{-1} + \frac{132}{\beta_1}\bar{C}_{l-1} + 10\right)
\end{aligned}
$$

where the second inequality comes from the fact that $g_l = \frac{2}{\beta_1}\epsilon_l^2\bar{C}_l$ and the third inequality comes from that fact that $\bar{C}_{l-1} \leq \sum_{s=1}^{l-1} C_s \leq 2\beta_1\epsilon_{l-1}^{-2}$.

Summing over all $L = \lceil \frac{1}{2}\log(n/\beta_1)\rceil$ number of epochs, we have that, for any $n$,

Query complexity

$$
\begin{aligned}
&\leq \sum_{l=1}^{L}\sum_{t\in\mathcal{I}_l} \mathbb{E}[Q_t] \\
&\leq 40\beta_1\theta^*(2R^* + 3\epsilon_{L-1} + g_{L-1})\left(4(R^*)^2\epsilon_L^{-2} + 12R^*\epsilon_L^{-1}\right) \\
&\quad + 40\beta_1\theta^*(2R^* + 3\epsilon_{L-1} + g_{L-1})L\left(\frac{132}{\beta_1}\bar{C}_{total} + 10\right) \\
&= 40\beta_1\theta^*(2R^* + 3\epsilon_{L-1} + g_{L-1})\left(4(R^*)^2\frac{n}{\beta_1} + 12R^*\sqrt{\frac{n}{\beta_1}} + 5\log(n/\beta_1)\right) \\
&\quad + 2450\theta^*(2R^* + 3\epsilon_{L-1} + g_{L-1})\log(n/\beta_1)\bar{C}_{total} \\
&= \theta^*(2R^* + 3\epsilon_{L-1} + g_{L-1})\left(160(R^*)^2 n + 480R^*\sqrt{n\beta_1} + 200\beta_1\log(n/\beta_1)\right) \\
&\quad + 2450\theta^*(2R^* + 3\epsilon_{L-1} + g_{L-1})\log(n/\beta_1)\bar{C}_{total} \\
&\leq \mathcal{O}\left(\theta^*(R^* + 3\sqrt{\frac{\beta_1}{n}} + \frac{\overline{C}_{total}}{n})\left((R^*)^2 n + \log(n/\beta_1)\right)\beta_1\right) \\
&\quad + \mathcal{O}\left(\theta^*(R^* + 3\sqrt{\frac{\beta_1}{n}} + \frac{\overline{C}_{total}}{n})\log(n/\beta_1)\bar{C}_{total}\right)
\end{aligned}
$$

where the last inequality comes from the following lower bound,

$$
3\epsilon_{L-1} + g_{L-1} = 3\epsilon_{L-1} + \frac{2}{\beta_1}\overline{C}_{total}\epsilon_{L-1}^2 \geq 3\sqrt{\frac{\beta_1}{n}} + \frac{2\overline{C}_{total}}{n}
$$

**Now we will deal with the correctness.** By Lemma E.8, we have

$$\Delta_{h_{out}} \leq \frac{3}{2}\hat{\Delta}_{h_{out}}^{L-1} + \frac{3}{2}\epsilon_{L-1} + 3g_{L-1}$$
$$\leq 3\epsilon_{L-1} + 3g_{L-1}$$
$$\leq 6\sqrt{\frac{2\beta_1}{n}} + 3g_{L-1}$$
$$\leq 6\sqrt{\frac{2\beta_1}{n}} + 24\frac{\bar{C}_{total}}{n}$$

where the second inequality comes from the definition of $h_{\text{out}}$ and $V_L^{L-1}$ and the third and last inequality is just by replacing the value of $\epsilon_{L-1}$ and $g_{L-1}$. **Finally, we can write this result in the $\varepsilon$-accuracy form.** Set $6\sqrt{\frac{2\beta_1}{n}} := \varepsilon$, we have $n = \frac{72\beta_1}{\varepsilon^2}$.

.