# OpenReview forum: "Corruption Robust Active Learning"
_NeurIPS.cc/2021/Conference — NeurIPS 2021 Poster_

### Official Review · Reviewer_EcVJ · 2021-07-08

**Rating:** 6
**Confidence:** 3

**Summary:**

The paper discusses binary classification when the labels can be corrupted by noise in an active learning scenario. In this direction the authors show that in a benign corruption setting the, a known algorithm, which the authors call RobustCAL, can achieve pretty-much the same label complexity as in a non-corrupted setting. However, the algorithm may fail in general and the authors propose a variation of this algorithm in order to be able to deal with the general case. This latter algorithm, though it requires more labels compared to RobustCAL, nevertheless it does come with good theoretical properties.


**Limitations And Societal Impact:**

The authors could be more explicit with a paragraph dedicated for this purpose to explain the limitations of the algorithm. For example, while the statistical question is resolved in a nice way, nevertheless, the algorithm does not appear to be computationally efficient. So, this is one example of the limitations that could be discussed. Also, one can compare further the label complexity between RobustCAL and CALruption.


**Main Review:**

The paper discusses an interesting problem in an interesting framework. Active learning, as well as robustness guarantees are sought for in many situations for classification and the content of this paper lies at the intersection of these two fields.

I have not seen the proofs in detail. However, to me there are a few things that are a little bit unclear even at a higher level. For example, the authors cite 3 papers about RobustCAL, but it appears that the name comes from the most recent citation and it is a bit unclear if all the details are the same across all three references.

As another example, I would expect a little bit broader description of the results that are related to corrupted labels. For example, the first paper that comes to one's mind is "Learning From Noisy Examples", by Angluin and Laird with constant misclassification rate, and another one is Sloan's "Four types of noise in data for PAC learning" which discusses malicious misclassification. None of these two papers is cited and I believe they should -- even if they discuss results in the non-agnostic case. Then one can make the jump to the agnostic case and cite and compare related literature and finally discuss everything in the context of the paper where active learning is also taken into account.

Another issue that the paper has is that the papers are not listed in alphabetical order and this makes it hard to find the paper one is looking for.

For the claim in lines 38-40, which is the appropriate reference?

Missing the reference (and a brief discussion) on the original CAL algorithm/paper in the related work section.

- line 152: is the upper bound by epsilon, an assumption of the misspecification model?
- Algorithm 1, line 7: the equality is perhaps set membership?
- Algorithm 1, line 12: what is $\hat{h}_{t+1}$?

I would also like to see more discussion of the proposed algorithm (CALruption) on the main text and get more explanations on the actual operations that take place in the various lines of the pseudocode. Similarly, I think the paper would also benefit if one could state simple versions of the bounds, in terms of label complexity. For example, the label complexity in line 244 (Corollary 5.1) seems quite daunting to someone who reads the paper for the first time.

Some typos:
- l 136: extend -> extended
- l 143: game -> label?
- l 157, equation: X -> x (lowercase in the subscript of the numerator)
- l 161: Is $\hat{R}_{[1, n]}$ defined somewhere?
- l 203: In the Section -> In Section
- l 220: hypothesis -> hypotheses (both occurrences on that line)

Overall, I think the paper should be above the acceptance threshold, but I would like to see some responses on the above.

============================================================

I am happy with the different points that have been raised in the reviews and the responses that we have received. Therefore, I am maintaining my recommendation for acceptance.

**Time Spent Reviewing:**

6

---

> ### Author Response · Authors · 2021-08-10
> **Response to Reviewer EcVJ**
>
>
> Q1: "For example, the authors cite 3 papers about RobustCAL, but it appears that the name comes from the most recent citation and it is a bit unclear if all the details are the same across all three references."
>
> The only difference between our modified RobustCAL algorithm and the original one is that, the elimination threshold in the original one only has the $\sqrt{\hat{\rho}/t} + 1/t$ terms while our algorithm has an extra $\hat{\rho}$ term. Because $O(\sqrt{\hat{\rho}/t}) \leq O(1/t + \hat{\rho})$, this modified RobustCAL can be regarded as a slightly modified version of the original one. But we think it is still worthwhile to show that such classical algorithm can already achieve good results in this benign setting, while in multi-armed bandit regret minimization, such a hard elimination algorithm usually fails. We will modify the discussions in Line 170-175 to make it clearer as well as make the reference to original
>
>
> Q2: "As another example, I would expect a little bit broader description of the results that are related to corrupted labels. ... Then one can make the jump to the agnostic case and cite and compare related literature and finally discuss everything in the context of the paper where active learning is also taken into account."
>
>  Thanks for your suggestion! We will add these relevant sources to give a more clear introduction on the noisy/corrupted label settings.
>
>
> Q3: "The authors could be more explicit with a paragraph dedicated for this purpose to explain the limitations of the algorithm. For example, while the statistical question is resolved in a nice way, nevertheless, the algorithm does not appear to be computationally efficient. So, this is one example of the limitations that could be discussed. Also, one can compare further the label complexity between RobustCAL and CALruption"
>
> Thanks for your suggestion, we will add a paragraph discussing the limitations of these algorithms. As a short response here:
>
> - The algorithm is not computationally efficient mainly because we need to loop over each hypothesis pair. But we believe that could be further improved using some existing techniques like in [Beygelzimer et al 2010, Huang et al 2015], as we comment in the response to Reviewer mLmE. Nevertheless, computational efficiency is not the main focus of this paper and we hope to improve it in the future.
> - For the label complexity between RobustCAL and CALruption, we discussed a little bit in line 263-267. That is, as long as the corruption satisfies the assumption in Theorem 4.1, RobustCAL will has a better label complexity than Calruption. But we will make it more clear in our next version
>
>
> Q4: "For the claim in lines 38-40, which is the appropriate reference?"
>
> This has been discussed in Section 3 Theorem 3.1, where we show that the passive learning has a corruption-related term $CR^*/n$, therefore, when $C^*$ is sublinear, this term will eventually go to 0 while in the active learning case you may forever eliminate the best hypothesis.
>
>
> Q5:  "line 152: is the upper bound by epsilon, an assumption of the misspecification model?"
>
> -   Yes, $\epsilon$ here measures the amount of misspecification / contamination. We will clarify it in the final version.
>
> Other minors:
>
> -  "Algorithm 1, line 7: the equality is perhaps set membership?"
>     Yes.
> -   "Algorithm 1, line 12: what is \hat{h}_t+1?"
>      This is a typo. We will delete that.
> - "Another issue that the paper has is that the papers are not listed in alphabetical order and this makes it hard to find the paper one is looking for"
>      Thanks for the reminder! We will fix that.

---

### Official Review · Reviewer_yion · 2021-07-14

**Rating:** 7
**Confidence:** 3

**Summary:**

The authors consider robust active learning. They demonstrate that an existing algorithm Robust CAL can be tricked into permanently discarding the optimal hypothesis by large amounts of corruption early on during learning (followed by no corruption to keep overall levels low). The authors repair this flaw by modifying the algorithm to do a soft-elimination so that no hypothesis is permanently discarded. However, the cost is a higher unlabeled sample complexity, which also creates a weakness to increasing corruption.


**Ethical Concerns:**

None.

**Limitations And Societal Impact:**

The authors address limitations of unlabeled sample complexity and other weak points of the proposed algorithm.

**Main Review:**

This work improves our understanding of the impact of corruption in active learning. While intuitative, it provides machinery for understanding the importance of the timing of the corruption during the learning process.

As pointed out by the authors, the proposed algorithm requires significantly more unlabeled samples. While it is often assumed that unlabeled data are cheap and/or easily obtained, this is not always the case.

Overall the paper is well written and clear. The only real criticism I have is that the result is of middling impact. By improving our understanding of robust active learning, the authors make a step toward better learning algorithms, but do not create fundamentally new capabilities. Thus, the impact is not low, but neither is it high.



**Time Spent Reviewing:**

3

---

> ### Author Response · Authors · 2021-08-10
> **Response to reviewer yion**
>
> Q1: "As pointed out by the authors, the proposed algorithm requires significantly more unlabeled samples. While it is often assumed that unlabeled data are cheap and/or easily obtained, this is not always the case."
>
> Yes, this is the limitation of the paper that we will try to solve in the future.

---

### Official Review · Reviewer_mLmE · 2021-07-16

**Rating:** 6
**Confidence:** 4

**Summary:**

This paper considers active learning with an "oblivious adversarial model": before the start of learning, an adversarial chooses a series of distributions $P_t(Y\mid X)$ , and at time t the label $Y$ would be drawn from that distribution. The adversarial level is characterized by $C_{total}=\sum_t \sup_x |P_t(Y\mid X=x)-P(Y\mid X=x)|$ where $P(Y\mid X=x)$ is the distribution over which the error for the model is calculated.

It first shows an error bound for passive learning with ERM. Then, it shows a slightly modified version of CAL (disagreement based active learning) could tolerate around $C_{total} <= n/8$ and achieve an error of $R^* + \epsilon + O(R^* C_{total}/n)$ where R* is the best error achievable without corruption and n is the number of unlabeled samples. Its label complexity is similar to that of the original CAL.

To tolerate any level of adversarial, instead of shrinking the version space permanently, it proposes a new algorithm that queries examples with some carefully designed probabilities. It gives its label complexity bounds, and shows that (1) its label complexity matches standard CAL without corruption; (2) it can tolerate higher corruption ($C_{total}>n/8$) while the previous modified CAL cannot; (3) its label complexity is slightly worse when the adversarial corruption is not high (e.g. the mispecification case where the probability that the adversarial corrupts the label in each round is uniformly upper bounded).

**Limitations And Societal Impact:**

See main review

**Main Review:**

This paper considers an interesting setting where the noise condition is harder than what existing active learning papers studied. Its results are new and nontrivial. I briefly went over its proofs and they look sound to me. However, this paper in its current shape is a bit premature to me, and I'm not sure if its quality is good enough for NeurIPS.

Main concerns:
- Clarity.
  - The main algorithm, Algorithm 2, is not very well explained. The high level explanations before Theorem 5.1 are nice, but it would be better if you can provide some explanation and *intuition* of detailed designs (if page limit is a problem, putting them in Appendix would still be helpful). For example:
    - how is the robust estimator constructed? What's the intuition behind the equation?
    - lines 9~13 in Algorithm 2 are quite hard to parse. Could you explain what these variables stand for? What are the rationale behind the chosen values?
    - If there is no corruption or under the mispecification case, is there any connection between Algorithm 2 and existing active learning algorithms with importance reweighting like [Beygelzimer et al 2010, Huang et al 2015] in your reference?
  - There are a lot of notations. It would be clearer if you can have a list of notations, even just in Appendix.
- Setup & results
  - The characterization of corruption level $C_{total}$ looks quite loose. One extreme example is that this quantity can be large even if the adversarial only corrupts the labels on a zero-measure set. It would make more sense if this measure takes the distribution of P(X) into account.
  - This is a relatively new setting, it would help understand the hardness of the problem and optimality of the proposed methods if you could show some lower bounds (even just for passive learning).
  - This paper only considers *oblivious* adversary model where the adversary has to decide its corrupted distribution beforehand. This looks a bit weak. What's the main obstacle to work against that?

Minor issues:
- It might be worth citing [1] though it only works under the mispecification case for linear separators.
- It would be clearer if you could provide some simple examples to show how traditional active learning algorithm fails with adversarial corruption.
- Algorithm 2 has a worse label complexity than modified CAL under low corruption.
- Disagreement coefficients are often not a constant (for example it can be O(\sqrt{d}) for linear classifiers with log-concave distributions).

A technical question:
At a glance, the guarantee for the robust estimator (Lemma D.2) is quite similar to what one typically gets with naive importance sampling. Could you state why importance sampling does not work for your use case?

[1] Awasthi, Pranjal, Maria Florina Balcan, and Philip M. Long. "The power of localization for efficiently learning linear separators with noise." Proceedings of the forty-sixth annual ACM symposium on Theory of computing. 2014.


**Time Spent Reviewing:**

8

---

> ### Author Response · Authors · 2021-08-10
> **Response to reviewer mLmE**
>
>
> Q1: " How is the robust estimator constructed? What's the intuition behind the equation?"
>
> The details of the construction of a robust estimator (Catoni estimator) is shown in Lemma D.1. The main motivation of using that estimator is that we want to estimate the true gap between each pair of hypotheses, but the simple empirical estimator will lead to potentially rare but large variance (line 227-228). Here this eqn.(1) shows the difference between the estimated one and the true one can be upper bounded by $\sqrt{\frac{\text{variance term}}{N}}$, where the $\frac{\hat{\rho}}{q}$ term is the variance term. (Note this is similar to the Bernstein-type concentration inequality.)
>
>
> Q2: "lines 9~13 in Algorithm 2 are quite hard to parse. Could you explain what these variables stand for? What are the rationale behind the chosen values?"
>
>  We hope our comments to all reviewers clarified things.
>
>
> Q3: "If there is no corruption or under the mispecification case, is there any connection between Algorithm 2 and existing active learning algorithms with importance reweighting like [Beygelzimer et al 2010, Huang et al 2015] in your reference?"
>
> A similarity of those papers with our own is the use of an importance weighted estimator. Besides that, other connections between our paper are shown as below:
>
> - For [Huang et al 2015], : If there is no corruption, the algorithm will be close to the Robust CAL algorithm. As we stated in the explanation to Line 12, the $V_{l+1}^l$ set is the active hypothesis set in the Robust CAL algorithm and the total label complexity from other sets than $V_{l+1}^l$ will be limited. [Huang et al 2015] propose an improved algorithm based on Robust CAL which minimizes the probability of querying inside the disagreement region, so we believe our algorithm may be able to adopt [Huang et al 2015]’s idea to further improve the efficiency. But overall this paper still uses the “hard elimination” methods. So fundamentally it is still quite different to our main contribution.
>
> - For [Beygelzimer et al 2010]: At a high level, this paper is more closely related to our ideas because it also uses a “soft version-space elimination” method. In the no corruption case, we believe our algorithm will behave similarly because in Line 13 of our algorithm, the query probability is roughly in proportional to the $1/\hat{\Delta}_h^l$ (although different in details). This is similar to the $P_k = O(1/G_k)$ statement in [Beygelzimer et al 2010]. Here $\Delta_h$ and $G_k$ are all close to the true gap between hypothesis $h(k)$ and the optimal one. But how to adopt the advantage of [Beygelzimer et al 2010] into the corruption case remains open.
>
>
> Q4: "The characterization of corruption level Ctotal looks quite loose. One extreme example is that this quantity can be large even if the adversarial only corrupts the labels on a zero-measure set. It would make more sense if this measure takes the distribution of P(X) into account."
>
> This is an excellent observation, but it is not clear that a metric weighted by the relative frequency of a particular x is possible to control. For instance, an active learning algorithm may find very rare x’s extremely informative and affect the model greatly if their labels are requested. But if they occur infrequently, such a weighted metric would suggest that very little corruption has occurred. Understanding what threat models can be controlled is an incredibly exciting research direction.
>
>
> Q5: "This is a relatively new setting, it would help understand the hardness of the problem and optimality of the proposed methods if you could show some lower bounds (even just for passive learning)."
>
> While we enthusiastically agree that this novel setting requires more attention and understanding, we tried to provide intuition about how standard active learning algorithms can fail if they are not robust to adversaries. In particular, we showed that elimination style algorithms can fail catastrophically with adversaries that mislead the algorithm at early stages.
>
> For the passive setting, we believe our Theorem 3.1's sample complexity ($n$) is tight as it matches the sample complexity in the uncorrupted case. We leave it as future direction to study whether the term C_{total} R^*/n is tight.
>
>
> Q6: "This paper only considers oblivious adversary model where the adversary has to decide its corrupted distribution beforehand. This looks a bit weak. What's the main obstacle to work against that."
>
> Our apologies! We hope our comments to all reviewers clarified this point. .

---

> > ### Comment · Reviewer_mLmE · 2021-08-14
> > **Response to Author Response**
> >
> > Thanks for the explanation! A few follow-up questions & comments:
> >
> > Q1:Naive importance sampling should also give something like $\sqrt{\frac{\text{variance term}}{N}}$ (and it is much simpler than the Catoni estimator). Why doesn't that work?
> >
> > Q2: The added explanation is indeed quite helpful. Two very minor comments:
> > - It might be worth mentioning that in line 9, the distribution D that the argmin is taken over has the same marginal distribution P(X) as D_t.
> > - It is not very clear what "policy set" means in your explanation for line 13.
> >
> > Q3: "If there is no corruption, the algorithm will be close to the Robust CAL algorithm." Actually this is not very obvious to me. In particular, I don't see how "the total label complexity from other sets than $V_{l+1}^l$ will be limited." Perhaps it would be clearer if you could provide a more detailed explanation in the future version.
> >
> > Q4: I don't think the difficulty ("for instance ...") you mentioned is really an issue. If an example x is very rare, then it would be unlikely your algorithm would get a sample on it, and even if it does, it shouldn't matter too much if the algorithm makes a mistake on it. It seems to me you just need to find some better definition of the corruption level properly, and your algorithm might be able to adapt to it with more refined analysis.

---

> > > ### Author Response · Authors · 2021-08-19
> > > **Further Response**
> > >
> > > Answer to Q1:
> > > We want to thanks for the reviewer to point this out and we realized that the naive importance sampling should work. Actually here we use the Catoni estimator because we start this project with the pool-based AL problem where the naive importance sampling might give you $\sqrt{d/T}+d/T$. This is not good when $d >> T$, which might be the case in pool-based AL. But we now realize that in the streaming setting presented in this paper that won't be a problem. So the line 8,9 in the main algorithm and its corresponding analysis can be simplified. But other parts of the algorithm and analysis are still necessary. We will check the details and update that in the next version.
> > >
> > > Answer to Q2:
> > > Thanks for your suggestions.
> > >
> > > Answer to Q3:
> > > We realized that saying  "be close to the Robust CAL algorithm" is a little bit confusing. Actually, I think it should be more close to the [Beygelzimer et al 2010] instead of elimination type algorithm. We will make it more clear in the paper. But on the high level, I just try to say they both try to contract the active version space.
> > >
> > > Answer To Q4:
> > > Yes we generally agree with you. We don't have a clear answer at this moment and will keep you update if we find a better definition.

---

### Official Review · Reviewer_Ev2r · 2021-07-20

**Rating:** 6
**Confidence:** 3

**Summary:**

The authors study the problem of active learning in binary classification with the presence of agnostic noise in the streaming model. The adversary chooses a priori some corrupted distributions $D_t$ for each round $t$. The goal is to design an algorithm that outputs a hypothesis $h$ that minimizes the expected risk using a few labeled samples. In this work, they assume that they do not have any a priori information on corruption. The authors proceed by first analyzing the ERM for the passive learning for using it as a benchmark for the rest of the algorithms. The authors analyze two algorithms the RobustCAL (which has been analyzed extensively in previous works) and the CALruption which is the new algorithm they propose. For RobustCAL they show that as long as the corruption is bounded enough (the number of corruptions is less than $t/8$) in all the $t=1,2,4,...,2^k$ then the classic RobustCAL along with a regularization term will output a hypothesis that achieves expected risk competitive with the passive learning, i.e., the RobustCAL will not eliminate the optimal hypothesis in any elimination step. Moreover, they show that the regularization term is necessary by showing that without this term the best hypothesis will be eliminated. In order to remove the assumption on the bounded corruptions, the authors suggest a new algorithm CALruption, that overcomes this issue. However, this algorithm in the general case outputs a hypothesis $h$, such that $R(h)-R^* \leq \epsilon +O(\bar{C}/n)$ where RobustCAL gets $R(h)-R^* \leq \epsilon +O(R^* C/n)$, $\bar{C}$ is the number of corruptions in each epoch with some weights if the corruptions are large or small. Furthermore, if the corruptions are bounded enough in each epoch, then the difference of expected risk can be improved to $\epsilon +O(R^*C/n)$.


**Limitations And Societal Impact:**

This is a theoretical work and has no negative societal impact.

**Main Review:**


### Main Review
- Strengths of this work:
1. Extends previous approaches on robust active learning to the agnostic streaming setting, when there is not a priori information on the corruption size.
2. Shows why known approaches fail in this setting.
3. Suggest a new approach for this setting that in several cases is optimal.

- Weaknesses:
1. The algorithm is much more complicated and less intuitive than RobustCAL.
2. The difference between $R^* C/n$ and $C/n$ seems a large gap.


### Other comments:
- The paper is well organized and well written except pages 6-7 (and Appendix) which are the main contributions, which also hold for the respective section in the appendix (section D).
- I would suggest the authors rephrase/clean algorithm 2 to make it easier to parse.
- This section is notation-heavy and the authors fail to explain the use of each symbol. In order to understand several parts/notations, I needed to check the proofs in Appendix.
- Appendix is full of typos and difficult to read.
- I suggest the authors provide a full proof in the appendix and provide an intuition for the algorithm and the proof.
Some Typos:
- In label complexity: $|H|^2$ should be $|H|$ because of $O$.
- Line 178: at least*
- 157: h should be $h^\ast$ and "and."
-  Typos in Appendix (including not limited to): Lines: 532, 558-559, 562-563,566-567,570,575-576,624.
- Some inequalities need justification.

I suggest the authors improve the clarity of this work in the camera-ready version (and in the full version of this work). For instance, I cannot understand why in some lemma statements the authors prove inequalities instead of doing it in the proof, e.g., Lemma C1.

**Time Spent Reviewing:**

4-5

---

> ### Author Response · Authors · 2021-08-10
> **Response to Reviewer Ev2r**
>
> Q1: "The algorithm is much more complicated and less intuitive than RobustCAL."
>
> We regret that the algorithm appears so complex, but this is a shared attribute of many robust bandit algorithms to overcome adversarial attacks. Algorithms based on the FTRL/OMD framework can be simpler, but it is unclear how to adapt these methods to active learning. This is a worthwhile research question.
>
>
> Q2: "The paper is well organized and well written except pages 6-7 (and Appendix) which are the main contributions, which also hold for the respective section in the appendix (section D)."
>
> Thank you for your suggestions--we hope our comments to all reviewers clarified things.
>
> Q3: "The difference between R∗C/n and C/n seems a large gap"
>
> This gap is indeed regrettable. We briefly discussed in lines 254-246, 260-262 where this gap gets introduced, but we have not yet identified a path around these difficulties.

---

### Author Response · Authors · 2021-08-10
**Comments to all reviewers**

We would like to begin by thanking all the reviewers for their thoughtful comments. We have responded to each of you individually after making a few remarks to the group.

We would like to correct one minor inaccuracy stated in our paper (i.e., replace a weaker guarantee with a stronger one). We stated that our main algorithm works for oblivious adversaries, but, in fact, it also works for adaptive adversaries, which are strictly more general. To be specific, the adversary can decide $\tilde{\eta}_t^x$ based on the previous history before $t$.

In addition, many reviewers suggest we refine the explanation of the main algorithm and its analysis. We will fix all the typos pointed out by reviewers and will make a notation table. Meanwhile, as suggested by Reviewer mLmE, we will provide a more detailed explanation on line 9~13.

---- In Line 9, we are going to estimate the underlying distribution of samples based on the collected samples. To be specific, we have the estimated gap between each pair of h and $h’$, so the initial desire is to find a proper distribution $\hat{\mathcal{D}}$ that induces all gaps uniformly close to all the estimated gaps. But this is impossible, so we instead choose the distribution that minimizes the worst-case pairs scaled with its variance. With such an estimated distribution, we can naturally get the estimated error of each hypothesis $h$ denoted as $R_{\hat{\mathcal{D}}}(h)$.

---- In Line 10, recall that we already have the $R_{\hat{\mathcal{D}}}(h)$, and the previously estimated gap between any hypothesis $h$ and the previous estimated best hypothesis $\hat{h}_*^{l-1}$, denoted as $\hat{\Delta}_h^{l-1}$.

So based on these two terms,  we can have a pessimistic estimation of the current best hypothesis $\hat{h}_*^l$.

---- Then in Line 11,  based on the estimated best hypothesis $\hat{h}_*^l$, we can further have a new estimated gap $\hat{\Delta}_h^{l}$.

---- Up to this point, we have an estimate of the performance of each hypothesis ( $\hat{\Delta}_h^{l}$). Now recall that in the traditional elimination-style algorithms like Robust CAL, we will permanently eliminate all the hypotheses for which  $\hat{\Delta}_h^{l} $ is larger than some threshold and then do a disagreement-based query on the remaining hypothesis set.  But here, the learner never makes a  ``hard"  decision to eliminate any hypothesis. Instead, it assigns different query probability to each $x$ based on the estimated gap $\hat{\Delta}_h^{l} $  for each hypothesis, That is what Line 12 and Line 13 are doing. To be specific:

---- In Line 12, we divide the hypothesis into $l+1$ sets based on $\hat{\Delta}_h^{l}$.

Again in the traditional elimination-style algorithm, the only remaining active hypothesis set is $V_{l+1}^l$.

---- In Line 13, based on these layered hypothesis sets, we are going to assign the query probability on the incoming $x$. Intuitively,  for each $x$, we want to find the smallest policy set it belongs to, among all those layered sets. Then, because the smaller the set is, the smaller its corresponding estimated gap is, so intuitively, we want to assign a higher query probability to those $x$ that have a smaller corresponding hypothesis set.

---

### Decision · Program_Chairs · 2021-09-27

**Decision:**

Accept (Poster)

**Comment:**

The paper studies active learning under label corruptions and presents a robust version of the classical CAL algorithm. All the reviewers agreed that the paper studies an interesting direction and presents good theoretical results. While the results in the paper do not present a complete picture of the problem setting, there is enough technical novelty to make the paper slightly above the bar for publication. The authors are encourage to carefully take into account the reviewers' comments when preparing the final version.